## Registered report

psychology

choice-consistency, memory, revealed preference, cognitive modelling

**Author for correspondence:**
Felix J. Nitsch
e-mail: felix.nitsch@hhu.de

# Influence of memory processes on choice-consistency

## Felix J. Nitsch and Tobias Kalenscher

Comparative Psychology, Heinrich-Heine-University Düsseldorf, 40225 Düsseldorf, Germany

 FJN, 0000-0002-7832-7498

Choice-consistency is considered a hallmark of rational value-based choice. However, because the cognitive apparatus supporting decision-making is imperfect, real decision-makers often show some degree of choice inconsistency. Cognitive models are necessary to complement idealized choice axioms with attention, perception and memory processes. Specifically, compelling theoretical work suggests that the (imperfect) retention of choice-relevant memories might be important for choice-consistency, but this hypothesis has not been tested directly. We used a novel multi-attribute visual choice paradigm to experimentally test the influence of memory retrieval of exemplars on choice-consistency. Our manipulation check confirmed that our retention interval manipulation successfully reduced memory representation strength. Given this, we found strong evidence against our hypothesis that choice-consistency decreases with increasing retention time. However, quality controls indicated that the choice-consistency of our participants was non-discernable from random behaviour. In addition, an exploratory analysis showed essentially no test–retest reliability of choice-consistency between two observations. Taken together, this suggests the presence of a floor effect in our data and, thus, low data quality for conclusively evaluating our hypotheses. Further exploration tentatively suggested a high difficulty of discriminating between the choice objects driving this floor effect.

## 1. Influence of memory processes on choice-consistency

Imagine a stock trader who wants to trade stocks on two different days and plans to invest a starting capital of 600€. On the first day, shares of company A cost 200€ and shares of company B 150€. The stock trader buys 3 shares of company A and 0 shares of company B on the first day. On the second day, the share price of company A sinks to 150€ and the share price of company B rises to 200€. How should the stock trader respond to such a volatile stock market?

A naïve (and inconsistent) stock trader might be tempted to prematurely sell the shares of company A and instead invest into company B. However, this would incur sensitive losses to the trader (de facto 150 €, a fourth of the starting capital). More importantly, continuously selling shares cheaper than buying them will inevitably lead to the loss of all capital and being driven out of the market (the so-called *money pump phenomenon*). Such investment behaviour might, for example, arise from an inconsistent company value definition. By contrast, a consistent stock trader would base trading decisions on financial analysis, for example considering liquidity, book-to-market value, degree of state-ownership and past performance. This would result in a more robust value definition of company shares than the share price on a given day. Such a stock trader would, ideally, buy stocks at low prices and sell stocks for a profit, using the price volatility advantageously.

Consistent choice can be formalized according to revealed preference theory [1–3]. It requires consistent integration of multiple-choice attributes [4] so that it can be rationalized by a monotonic concave utility function [5]. In the example above, a utility could be given by the consistent integration of liquidity, book-to-market value, degree of state-ownership and past performance of company shares. Formally, revealed preference theory in its generalized form can be defined as a bound on the structure of the preference relation. Varian [6] provides a summary of revealed preference theory.

In practice, revealed preference theory is often violated by seemingly inconsistent choice, leading to some researchers proposing sensible relaxations of the choice axioms [7] or the abandonment of choice axioms altogether in favour of a variety of heuristics [8].

An important requirement for consistent choice according to revealed preference theory is the stability of preferences and goal structures. The decision maker must have 'a definitive structure of wants' [5]. While the stability of preferences over prolonged time spans is considered trivial by some [9], others pointed out that preferences may change by endogenous and exogenous cause [10]. Importantly, such dynamic changes of preferences can be the result of natural psychological processes such as attentional shifts, memory encoding and retrieval. Query Theory [11] proposes that preferences are not always directly accessible to or completely defined by the decision maker. Instead, relevant experiences with the choice options are retrieved from memory to construct preferences during the decision process: 'Preferences, like all knowledge, are subject to the processes and dynamics associated with retrieval from memory' [12]. Gabaix & Laibson [13] propose in a similar notion that value-based choices are guided by imperfect Bayesian forecasting of future values. These forecasts are derived from prior beliefs and previous experiences. Failure of sufficient retrieval of such memories could result in unstable and incompletely defined preferences and, thus, choice inconsistency. Congruently with Query Theory, recent neuropsychological research finds evidence for the relevance of memory-related structures for value-based choices [14].

A problem of such choice-relevant memory failures is that they are not directly observable from behaviour: we can neither assess which nor how well choice-relevant memories are retrieved from choice behaviour. In a recent preregistered study, Levin *et al.* [15] offered a trait heterogeneity-based approach to the problem. The authors recruited people who were at least 65 years old to test for the effect of differences in memory abilities (measured by a cognitive assessment battery) on inconsistency in food choice. Participants rated a catalogue of food items on a Likert scale. Afterwards, they made repeated pair-wise choices between all possible pairs of food items from the catalogue. Memory ability heterogeneity affected the divergence of food ratings and actual choices. That is, participants with worse memory ability tended to more frequently choose items with a lower rating over items with a higher rating. However, unexpectedly, memory ability did not influence the transitivity of choice itself. It is important to note that the study by Levin *et al.* [15] did not offer any direct measurements of choice-relevant memory retrieval and deploys a non-experimental research design. Therefore, the process of how memory retrieval of goals and preferences affects choice-consistency remains unclear.

In the following sections, we will argue that the multi-attribute visual choice (MAVC) paradigm is a better-suited paradigm to assess the influence of memory retrieval of goals on choice-consistency. MAVC describes the comparative judgement of visual objects that are characterized by multiple attributes, e.g. orientation, colour and shape. Further, we will argue how the revealed preference framework allows a broader evaluation of choice-consistency than traditional accuracy measures of perceptual decisions.

## 1.1. Multi-attribute visual choice as a model of value-based choices

In our interpretation, the decision process as postulated by Query Theory [11] proposes, at the core, that information about the choice goals is retrieved from memory. Choice options are then compared along

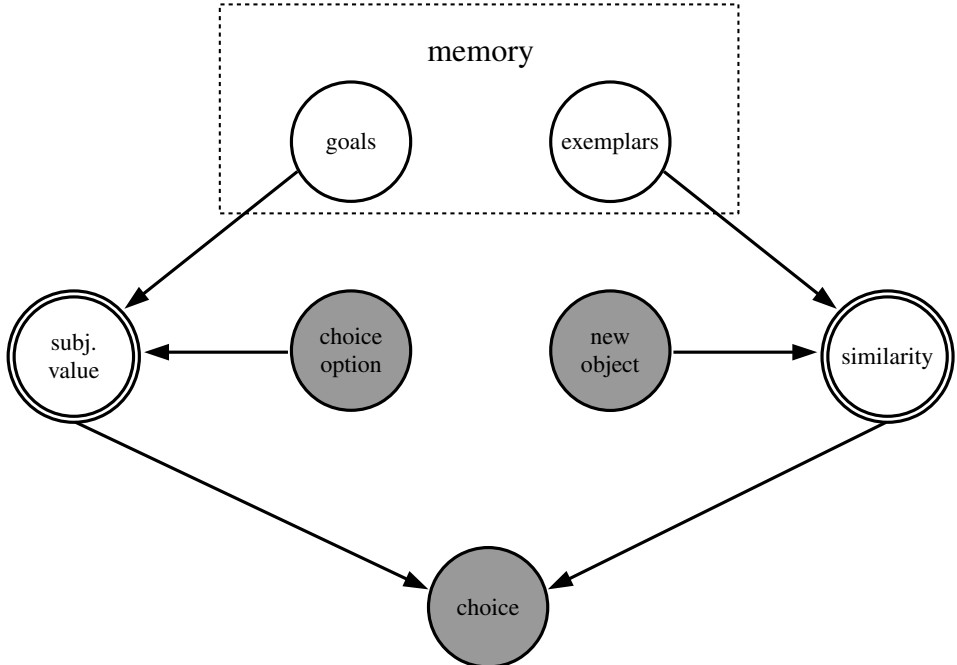

**Figure 1.** Mapping of concepts from GCM of categorization to Query Theory. Note. Simplified graphical representation of the decision process in Query Theory [11] and the GCM of categorization [18]. Nodes with dark background represent observed variables and nodes with white background represent latent variables. Nodes with single-line borders represent stochastic variables and nodes with double-line borders represent deterministic variables.

with all relevant dimensions to the choice goals and the option with maximum integrated goal similarity is chosen. In management science, these goals are also called performance targets [16].

For example, when a stock trader decides whether to invest in shares of company A or company B, the trader compares them regarding liquidity, book-to-market value, degree of state-ownership, past performance etc. to a benchmark of what they consider a good investment. The investment option in the choice set which comes closest to the benchmark in memory, or so-called choice goals, is chosen (assuming that not investing is no option).

This process is strikingly similar to the decision process proposed by the generalized context model (GCM) of categorization [17,18] in sensory perception, according to which new objects are perceptually categorized based on their similarity to stored exemplars. Exemplars are represented as points in a multi-attribute psychological space. Categorization then is performed by integrating the distance of the new object to the exemplars of a category in memory among all dimensions. Figure 1 shows how important concepts of value-based choice map onto equivalent concepts of MAVC. We propose that the process of comparing choice options to performance goals in value-based choice and to exemplars in MAVC is, psychologically, sufficiently similar to use MAVC as a model for value-based choice.

In MAVC, participants have to choose one out of a set of objects subjectively most similar to a previously learnt exemplar. The choice set objects vary in their similarity to the exemplar regarding multiple attributes. Such choices are comparable to, although not identical with, delayed-match-to-sample tasks [19–24].

An important advantage of MAVC tasks over value-based choice tasks is that we can experimentally induce and manipulate exemplar representations in MAVC, whereas goal representations are usually pre-existing, unknown and difficult to manipulate in value-based choice. We can, for example, experimentally manipulate memory representation strength of exemplars through changes of the retention interval (RI) between exemplar presentation and choice. These processes are well-studied and several off-the-shelf models for the relationship of memory representation strength and RI exist, e.g. exponential and power models [25].

## 1.2. Hypotheses

Based on the predictions of Query Theory [11] and neuropsychological evidence on the role of memory for value-based choice [14], we expect choice-consistency to be compromised when memory-based goal

representations are weak. Correlational evidence partly suggests that this is the case [15]; however, a direct experimental test of the relationship of the strength of memory-based goal representations and choice-consistency is missing and non-trivial to implement.

Based on theoretical considerations [4,17,18], we propose that MAVC can serve as a model for value-based choice. In MAVC, we can experimentally manipulate memory representation strength of exemplars through changes of the RI between exemplar presentation and choice. This maps to a manipulation of the strength of memory-based goal representations in our framework (figure 1). Revealed preference theory [1–3] can be used to analyse MAVC consistency without requiring assumptions about attribute weights or the parametric form of an integration function. Therefore, revealed preference theory can provide a general test of adherence to multi-attribute integration as formulated by the GCM of categorization and multi-attribute utility theory. We propose the following hypothesis:

> H1: As memory representations of exemplars are integral for MAVCs [18], we expect choice-consistency to decrease for longer RIs. That is, we expect an inverse relationship of RI between learning of the exemplar and choice and choice-consistency across multiple choices. Hence, we will provide experimental evidence on the role of memory representation strength of goals for choice-consistency.

Previous research on the retention of information shows that forgetting curves are nonlinear [25]. As we expect choice-consistency to be directly affected by the memory representation, we also expect the relationship of the RI and choice-consistency to be nonlinear.

> H2: We expect choice-consistency to decrease exponentially for longer retention intervals. That is, we expect an exponential model of the relationship of RI and choice-consistency to be more strongly supported by the data than a null model (predicting a truncated normal distribution around the mean of the data). The evidence on H2 will help us to quantify the role of memory representation strength of goals for choice-consistency beyond a directional prediction.
>
> H3: In congruence with H2, we expect the exponential decrease of choice-consistency for longer RIs to directly replicate in a new dataset. This is important, as replicability is a minimal requirement on the meaningfulness of a psychological phenomenon.

# 2. Methods

## 2.1. Why is revealed preference theory necessary?

Our main dependent variable is consistency in MAVC. We quantified visual choice-consistency with analysis tools borrowed from revealed preference theory. These are preferable over standard indices used in the visual memory and perception literature for conceptual and methodological reasons, as explained in the following.

Value-based choices usually involve trade-offs of different choice attributes. For example, a customer buying snacks might consider both taste and healthiness. While a chocolate bar is arguably tastier, an orange is healthier. A decision, therefore, requires integrating both choice attributes. Whether taste or health is given more weight is subjective. Concludingly, there is no objectively correct choice. A model of value-based choices should, therefore, include similar attribute trade-offs.

In MAVCs, the choice set stimuli represent a trade-off of similarity to the exemplar regarding multiple attributes. This means that, unlike in traditional memory recognition tasks, such as delayed-match-to-sample tasks, no visual object in the choice set is most similar to the exemplar with regard to all attributes. For example, consider a three-dimensional exemplar cube whose orientation is tilted along the X- and Y-axes (figure 2). One object in the choice set might be most similar to the exemplar regarding X-orientation while another one is similar regarding Y-orientation. Therefore, there is no objectively correct or dominating choice. This prohibits the use of traditional accuracy measures of perceptual choice that require a normatively correct choice option. By contrast, revealed preference theory allows one to test choice-consistency in the context of attribute trade-offs without making unnecessary assumptions about attribute weights or the form of an integration function [26].

## 2.2. Sample characteristics and exclusion criteria

Participants were recruited from undergraduate psychology students at Heinrich-Heine-University Düsseldorf, Germany on campus and by online adverts. Participants were at least 17 years old, had a

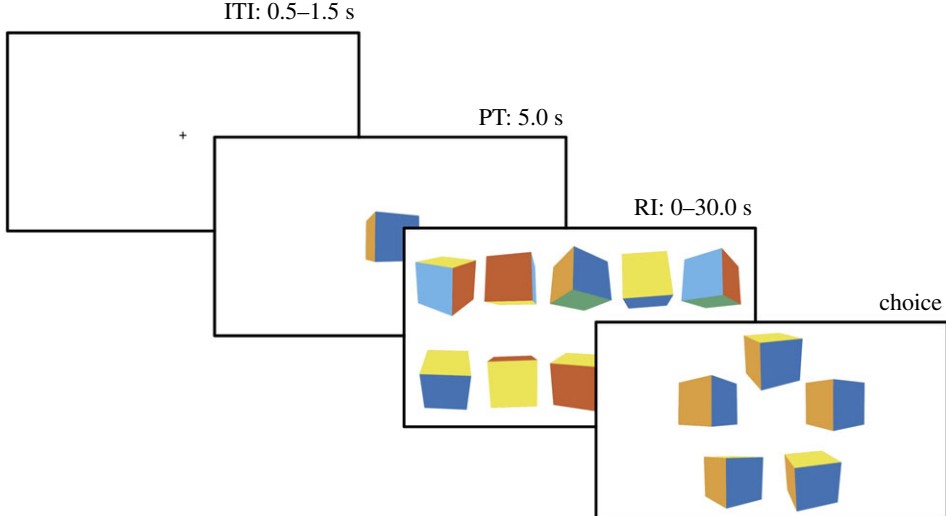

**Figure 2.** Timeline of a single choice trial. Note. From top left to bottom right: 1. The inter-trial interval (ITI) lasted 0.5 to 1.5 s. A fixation cross was presented in the middle of the screen. 2. During the presentation time (PT), an exemplar cube was presented for 5 s in the middle of the screen. 3. During the RI, a mask of cubes in random orientations was presented. The RI was randomly selected from an interval 0 to 30 s and was fixed per participant per block. 4. After the retention interval, the choice set of five cubes with different orientations was presented. Each element of the choice set was presented equidistantly around the exemplar. The order of the choice set elements was randomized. Participants had to make a forced a choice on which among the choice set stimuli is most similar in its orientation to the exemplar.

normal or corrected vision, a good level of German, no neuropsychological or psychiatric diseases and gave informed written consent. The study was approved by the local institutional review board of Heinrich-Heine-University and was conducted in accordance with the Declaration of Helsinki. Participants were reimbursed by course credit.

Participants were excluded from the analysis if they did not complete the full experimental session. We did not exclude partial data.

## 2.3. Experimental set-up and procedure

After participants had given their informed written consent, we assessed age, gender and mother-tongue.

Participants then solved a memory-based visual decision task (figure 2). In each trial, a three-dimensional exemplar cube was presented for 5 s.[1] Each side of the cube was characterized by a unique colour in the RGB space[2] from a colour scale optimized for colour-blind people [27]. Each side of the cube was 200px long. The exemplar cube had an orientation of 10, 75, 120, 185, 250 or 315 degrees on the $X$- and $Y$-axis and an orientation of 0 degrees on the $Z$-axis. After the presentation of the cube, a mask of 10 similar cubes (with random $X$- and $Y$-orientations) was presented to the participants for a short RI. After the RI, a choice set of five cubes with variable $X$- and $Y$-orientations was presented, and participants had to select one of the five cubes that had the most similar overall orientation to the exemplar.

The general notion of the task can be compared to that of delayed-match-to-sample tasks [19–24] with the difference that there is never a perfect match to the sample. Instead, the choice set stimuli represented a variable trade-off of orientation similarity to the exemplar regarding the $X$- and $Y$-axis. For example, a particular stimulus from the choice set might have had a similar $X$-orientation but a different $Y$-orientation. Another stimulus might have had a different $X$-orientation but a similar $Y$-orientation. Additionally, there could be trials where the choice set stimuli orientations resembled the exemplar orientation more closely and other trials where all choice set stimuli were quite differently oriented from the target stimuli.

---

[1]We chose this particular presentation time based on a pilot study (see section Pilot Experiment).

[2]RGB coordinates for each side of the cube. Front: (230, 159, 0). Back: (86, 180, 233). Bottom: (0, 158, 115). Top: (240, 228, 66). Right: (213, 94, 0). Left: (0, 114, 178).

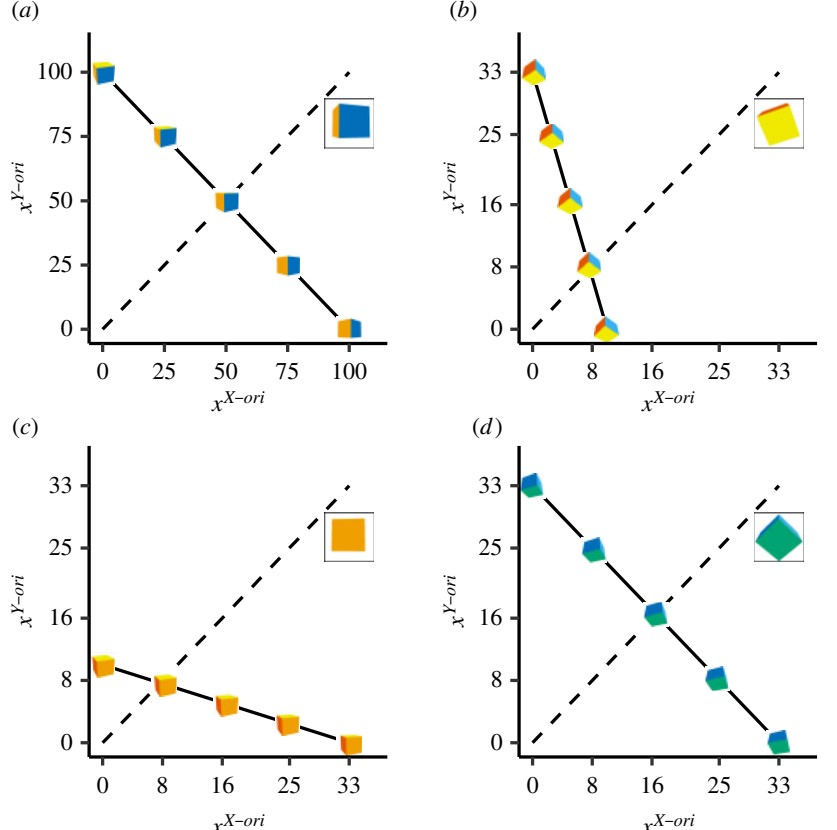

**Figure 3.** Construction of choice set of multi-attribute visual decision task from budget and prices. Note. Choice sets for four different exemplars and sets of prices. Exemplars are shown for each example in the square in the upper right corner of each panel. From (a–d) top left to bottom right: $p_1 = (1, 1)$, $p_2 = (3, 10)$, $p_3 = (10, 3)$, $p_4 = (3, 3)$. The size of the budget (set to $m = 100$) relative to the prices determines how similar the choice set stimuli are oriented to the exemplar overall. Hence, the choice set stimuli in (a) are overall more similarly oriented to their respective exemplar than the choice set stimuli in (d). The price ratio of the attributes determines the trade-off ratio of the X- and Y-orientation. Hence, the choice set stimuli in (b) are generally more similarly oriented to their respective exemplar along the Y-axis and less similarly oriented along the X-axis compared to (c) and vice versa. Axiomatic choice theory proposes that subjective similarity increases as a function of how far a choice object is located to (b) (indicated by the dashed line).

The task of the participants was, therefore, to mentally rotate each stimulus of the choice set until it matches the previously shown exemplar and evaluate which of the stimuli required the least mental rotation overall.

Framed in terms of revealed preference theory, each choice trial $i$ was constructed from a budget of $m = 100$ tokens. A pair of prices $p_i = (p_i^{X-ori}, p_i^{X-ori})$ was chosen uniform randomly from a numeric range of 1 to 3 and 1 to 10. The ranges were assigned to the prices randomly for each trial.

In value-based choice, the price of a good is the cost required to obtain a unit of this good. The budget line then constitutes all combinations of goods affordable spending a fixed budget. Thus, prices and budgets lines are constraints that restrict the possible choice set of combinations of goods out of all available goods. Similarly, the prices and budgets in our multi-attribute choice task constrained the choice set of visual objects out of all possible visual objects characterized by specific attribute values (figure 3). Given a fixed budget, the prices determined how much 'similarity' to the exemplar a participant could 'purchase' along with a given orientation axis. The 'cheaper' a given dimension, the more similarity to the exemplar on that dimension a participant could afford.

The five visual objects were then generated as equidistant points covering the entire budget line

$$x_i^{Y-ori} = \frac{m}{p_i^{Y-ori}} - x_i^{X-ori} \times p_i^{X-ori}/p_i^{Y-ori}.$$

Consequently, the choice set always included the extreme objects $x_{i,0} = (m/p_i^{X-ori}, 0)$ and $x_{i,m} = (0, m/p_i^{Y-ori})$. An attribute value of $x_i^{X-ori} = 0$ corresponded to an orientation difference of 30

degrees to the exemplar along the $X$-axis. With increased values of the $X$-orientation attribute, the choice object was turned towards the exemplar position along the $X$-axis. A single unit size amounted to 0.3 degree turn. An attribute value of $x_i^{X-ori} = 100$ corresponded to matching orientation to the exemplar along the $X$-axis. Likewise, an attribute value of $x_i^{Y-ori} = 0$ corresponded to an orientation difference of 30 degrees to the exemplar along the $Y$-axis. With increased values of the $Y$-orientation attribute, the choice object was turned towards the exemplar position along the $Y$-axis. A single unit size amounted to 0.3 degree turn. An attribute value of $x_i^{y-ori} = 100$ corresponded to matching orientation to the exemplar.

Participants received in-depth instructions about the task (see appendix). Further, they were presented with an animated rotating cube to familiarize with the cube itself.[3] Participants then first solved a practice block of 10 trials with a 1 s RI. This practice block served for the participants to familiarize with the design. Choices from the practice block were not included in the analysis. After the practice block, participants were asked to turn to the experimenter in case of questions. Then they solved two consecutive blocks of 20 trials each. For each test block, each participant was assigned a uniform random RI between 0 and 30 s (please refer to paragraph 'Floor and ceiling effects' below for a discussion of the optimal interval length). In total, participants made 20 decisions each for two distinct RIs.

After participants had completed the second test block, they solved a similar exemplar reconstruction task as quality control and manipulation check. The first three screens of each trial were equivalent to the procedure of the main task. Each trial started with the presentation of a fixation cross. Next, participants were presented with an exemplar cube with a certain orientation along the $X$- and $Y$-axis for 5 s. Then, participants were presented with a mask of 10 randomly oriented cubes (figure 2) for a certain RI. Each participant was assigned a RI of either 1, 5, 10 or 30 s for the memory reconstruction task. After the RI, participants were again presented with a single cube similar to the exemplar. The cube randomly matched the exemplar regarding either the $X$- or the $Y$-orientation, while the initial complementary orientation was chosen uniform randomly. Participants then had to turn the cube on the screen to match the exemplar regarding the complementary orientation using the arrow keys on the keyboard. Importantly, it was unknown to the participants whether they would have to reconstruct the $X$- or $Y$-orientation both during the presentation time and the RI. Participants solved 50 trials of the reconstruction task. Importantly, we did not use the results from the reconstruction task for our main analyses but as a manipulation check.

After completion of the reconstruction task, participants were debriefed about the goals of the study in written form and reimbursed via course credit.

The experimental task was presented with jsPsych [28]. All stimuli were presented on a Lenovo ThinkPad T590 laptop. Subjects were seated 30 cm away from the monitor in a dimly lighted room.

## 2.4. Revealed preference theory for multi-attribute visual choice

We measured consistency in MAVC, i.e. the degree of consistency in weighting the two orientation dimensions when comparing the memorized exemplar with the choice set. A participant would act consistent, for example, if they assigned more weight to orientation similarity to the exemplar along with one axis when it was expensive, and less weight when it was cheap. Revealed preference theory can be used to quantify the level of inconsistency in weighting the visual attributes in a straight forward manner.

Let $N \in \mathbb{N}$ be the number of different attributes of a visual object.

Following Nosofsky [18], let $X = \mathbb{R}_+^N$ be the non-negative, $N$-dimensional space of visual objects. Let $P = \mathbb{R}_+^N$ be the non-negative, $N$-dimensional space of prices of attribute similarities to the exemplar. Let $M = \mathbb{R}_+$ be the non-negative, one-dimensional space of budgets. Let $I = i, j \ldots \in \mathbb{N}$ denote observations of choice.

Let $x_i \in X$ be the chosen visual object of an observation $i \in I$. Each visual object $x_i$ is a $N$-dimensional vector of the shape $x_i = (x_i^1, x_i^2, \ldots, x_i^N)$, with each scalar component $x_i^n$ representing the similarity of the visual object $x_i$ with regard to attribute $n$.

Let $p_i \in P$ be the given prices of attribute similarities of an observation $i \in I$. Each prices $p_i$ are a $N$-dimensional vector of the shape $p_i = (p_i^1, p_i^2, \ldots, p_i^N)$, with each scalar component $p_i^n$ representing the price of similarity to the exemplar with regard to attribute $n$ per unit size.

Then, the scalar product $x_i p_j$ represents the total price of a visual object $x_i$ at some prices $p_j$. Let $m_i \in M$ be the given budget of an observation $i \in I$. We assume, that a decision maker spends all her budget so that $x_i p_i = m_i \; \forall i \in I$.

---

[3]For an impression visit: https://fjnitsch.github.io/files/html/Rotating_Cube.html

*Definition 1* (Direct Revealed Visual Preference). A visual object $x_i$ is directly revealed preferred to another visual object $x_j$ if and only if $x_j p_i \leq m_i$ and $x_i \neq x_j$. Then, we denote $x_i R_D x_j$.

*Definition 2* (Revealed Visual Preference). A visual object $x_i$ is revealed preferred to another visual object $x_j$ if there exists a transitive preference relation $x_i R_D x_k, x_k R_D x_l \ldots x_m R_D x_n, x_n R_D x_j$ between both bundles. We denote $x_i R x_j$. $R$ is the transitive closure of $R_D$.

*Definition 3* (Strict Direct Revealed Visual Preference). A visual object $x_i$ is strictly directly revealed preferred to another visual object $x_j$ if and only if $x_j p_i < m_i$. Then, we denote $x_i P_D x_j$.

*Axiom 1* (Generalized Axiom of Revealed Visual Preference). $x_i R \; x_j \rightarrow \neg(x_j P_D x_i) \; \forall i,j \in I$.

*Axiom 1* allows us to directly test multi-attribute perceptual choices for consistency. It is a necessary and sufficient condition for the choices to be rationalized by a monotonic concave attribute integration function and, thus, adherence to the GCM of categorization. If the choice data pass Axiom 1, this means that choices are made as if integrated subjective similarity to the exemplar is a function of objective similarity along with each attribute dimension (figure 3). A simple example of such an integration function could be that subjective similarity is the weighed sum of the similarity along with each attribute dimension. As one anonymous reviewer correctly pointed out, mental rotation may not necessarily be performed in an independent, piecewise fashion but possibly also in a holistic mode [29], at least for some participants [30]. We want to emphasize that any concave monotonic similarity function is consistent with revealed preference theory. Hence, an independent (i.e. additive) treatment of the two rotation axes is not required for our model.

However, contrary to Nosofsky [17], we do not need to make assumptions regarding the parametric form of such an integration function. Conversely, if the data do not pass Axiom 1 no GCM-style integration function of any monotonic concave specification can rationalize the data.

## 2.5. Preprocessing

For each test block and participant, we calculated the critical cost-efficiency index (CCEI) [5,31,32]. The CCEI can be interpreted as how consistently multiple attributes of choice options are integrated into a decision value. The CCEI denotes the 'amount by which each budget constraint must be adjusted in order to remove all violations of GARP' [33, p. 1927]. Computationally, the CCEI presents a relaxation of Axiom 1, so that only $x_i R \; x_j \rightarrow \neg(x_j p_j \times \text{CCEI} > x_i p_j) \; \forall i,j \in I$ must hold. It ranges from zero to one. A value of one denotes perfect consistency: the attributes are weighed consistently across all choices. The CCEI approaches zero as choices become increasingly inconsistent, which means that choice option attributes are weighed inconsistently across different trials. The CCEI is the most common indicator of compliance with choice-consistency as defined by revealed preference theory and has been applied in value-based choice in various domains [34]. Further, we explored the robustness of our results using similar indices such as the money pump index [35], the Houtman–Maks index [36] and the minimum cost index [37]. However, since all of these metrics measure slightly different constructs, we restrained our preregistered analysis to the CCEI.

## 2.6. Analysis pipeline

Per participant, the data from one test block were randomly selected for testing for an inverse relationship of RI and choice-consistency, Bayes factor model comparison and parameter estimation. We call these data *training set*. The other test block was used to replicate our results in a new dataset. Therefore, these data were not used for other analyses. We call these data *test set*.

For all analyses, we used a Bayesian framework of inference. Bayesian statistics allows us to express confidence that a parameter is within a certain range, to extend parameter estimation naturally for complicated models, to express evidence for or against hypotheses on a continuous scale and to monitor evidence accumulation [38].

All our analyses were conducted in RStudio [39]. We used the following R packages: BayesFactor [40], runjags [41], Tidyverse [42] and patchwork [43]. Further, we used the JAGS software [44] for the analysis of Bayesian graphical models.

### 2.6.1. H1: test for an inverse relationship of retention interval and choice-consistency

In order to test for an inverse relationship of RI and choice-consistency, we calculated Kendall's Tau in the training set. Compared to Pearson's *r*, it is robust to outliers and violations of normality and expresses dependence in terms of monotonicity instead of linearity [45]. This is important, as we neither

expected choice-consistency (index ranging from 0 to 1) nor the RI (uniformly sampled from an interval of 0 to 30 s) to be normally distributed, nor both variables to have a linear relationship. We followed the exact procedure proposed by van Doorn *et al.* [45] to test for an inverse relationship of RI and choice-consistency using Bayes factor analysis for Kendall's Tau.

### 2.6.2. H2: Bayes factor model comparison of exponential and null model

In order to gain further insights into the relationship of RI and choice-consistency, we planned to test which model is supported more strongly by the data of the training set (but see section Interpretative Plan and results for H1 as to why we did not proceed to test this hypothesis). For this, we planned to use Bayes factor model comparison via the product space method [46]. We planned to test two candidate models against each other, which are specified in the following sections. We assumed both models to have equal prior probabilities.

$$p_{M1} = p_{M2} = 0.5.$$

The first candidate is inspired by forgetting models of item recall [25]; the second model is a null model assuming no effect of the RI on choice-consistency. They give rise to observed participant choice-consistency, given a RI.

Both candidate models are of the general form

$$CCEI_t \sim \text{Normal}(\mu_t, \sigma),$$

with $CCEI_t$, $\sigma \in [0, 1]$. $CCEI$ denotes the CCEI of a participant for one test block, and $t$ denotes the assigned RI for that block (ranging from 0 to 30 s). $\sigma$ accounts for random noise in the data. We assume all parameter values for $\sigma$ to be equally likely *a priori*.

$$\sigma \sim Beta(1,1).$$

$\mu_t$ denotes the expected choice-consistency given a RI and is specific to the model candidates.

#### 2.6.2.1. Exponential model

The first candidate model assumes that choice-consistency decreases exponentially with retention time. This means that the decreasing rate of consistency is constant over retention time. Following Averell & Heathcote [25], the function can be formalized in the following way:

$$\mu_t = a + (1 - a) \times b \times e^{-\alpha \times t}.$$

The parameter $a \in [0, 1]$ determines an asymptotical minimum level of choice-consistency after an infinite RI. The parameter $b \in [0, 1]$ determines choice-consistency at $t = 0$, which allows for imperfect choice-consistency unconditional on time-dependent processes when $b < 1$. The parameter $\alpha \in [0, 1]$ determines the retention time-constant decreasing rate of consistency. We assume that all parameter values are equally likely *a priori*.

Figure 4 displays a graphical representation of the model including prior specifications for all parameters.

#### 2.6.2.2. Null model

The second candidate model assumes that choice-consistency does not decrease as a function of the RI. The expected value of the choice-consistency distribution is, therefore, a constant:

$$\mu_t = c.$$

The parameter $c \in [0, 1]$ determines the expected value of the choice-consistency. We assume that all parameter values for $c$ are equally likely *a priori*:

$$c \sim Beta(1,1).$$

### 2.6.3. H3: replication for the test set

In order to test whether the relative advantage in support by the data for the exponential model in comparison to the null model replicates to a new dataset, we planned to obtain the replication Bayes

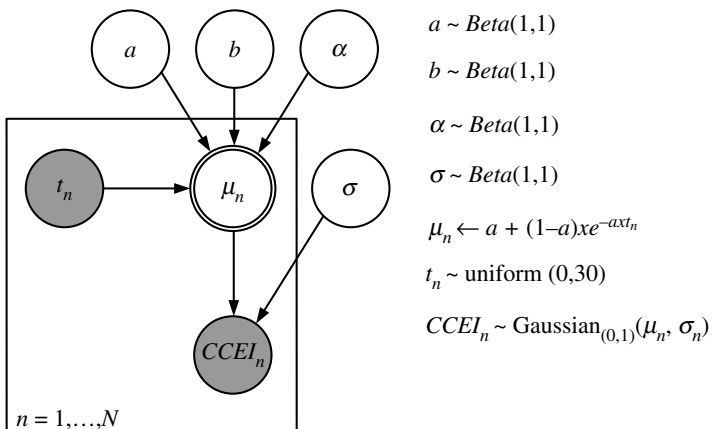

$a \sim Beta(1,1)$

$b \sim Beta(1,1)$

$\alpha \sim Beta(1,1)$

$\sigma \sim Beta(1,1)$

$\mu_n \leftarrow a + (1-a)xe^{-axt_n}$

$t_n \sim \text{uniform}\ (0,30)$

$CCEI_n \sim \text{Gaussian}_{(0,1)}(\mu_n, \sigma_n)$

**Figure 4.** Graphical exponential model of the relationship of retention interval and choice-consistency. Note. $n \in N$ denotes a single data point corresponding to a test block of a particular participant. $t_n$ denotes the retention interval of a given observation. $\mu_n$ denotes the expected choice-consistency of a given observation. $CCEI_n$ denotes the observed choice-consistency of a given observation. The parameter $a$ determines an asymptotical minimum level of choice-consistency after an infinite RI. The parameter $b$ determines choice-consistency at $t = 0$, which allows for imperfect choice-consistency unconditional on time-dependent processes when $b < 1$. The parameter $\alpha$ determines the constant decreasing rate of consistency. We assumed that all parameter values are equally likely a priori. Nodes with dark background represent observed variables, and nodes with white background represent latent variables. Nodes with single-line borders represent stochastic variables, and nodes with double-line borders represent deterministic variables.

factor using the held out test set using the method described by Ly et al. ([47]; but see section Interpretative Plan and results for H1 as to why we did not proceed to test this hypothesis). The replication Bayes factor is given by Bayes factor for the coerced dataset divided by the Bayes factor for the training set (obtained for H2):

$$BF_{10}(d_{\text{test}}|d_{\text{train}}) = \frac{BF_{10}(d_{\text{test}}, d_{\text{train}})}{BF_{10}(d_{\text{train}})}.$$

This evidence updating method does not require approximations and is especially useful for complex models as in our application case.

## 2.7. Interpretative plan

We followed the usual framework [48] for interpreting Bayes factors, which means that we considered a Bayes factor of $BF \geq 10$ as strong evidence for a hypothesis. Table 1 summarizes the interpretative plan for all hypotheses.

We collected further data until we reached a conclusive result for all hypotheses.

H1: should we find strong support for an inverse relationship of RI and choice-consistency, we would conclude that choice-consistency in MAVC depends on the memory representation strength of exemplars. Should we find strong evidence against an inverse relationship of RI and choice-consistency, this would question the role of memory representation of exemplars in MAVC. It could be concluded that choice-consistency is robust to indefinite goal representations. In this case, we would not proceed to test H2 and H3.

H2: should we find strong evidence, that the exponential model of the relationship of RI and choice-consistency is supported more strongly by the data than the null model, we would interpret this as preliminary evidence for the validity of the exponential model. However, a definitive interpretation would require the generalizability of the results for the test set. Furthermore, our statistical tests would only collect relative evidence for one model over another. It would still be possible that the true model is outside our model space. Therefore, careful inspection of the visualizations of the model predictions would be required (see figures 5 and 6). Should we find strong evidence in support of the null model, this would question the validity of an exponential model specifically, given positive evidence for H1. Again, a definitive interpretation would require the generalizability of the results for the test set.

H3: should we find strong evidence that the relative advantage in support by the data for the exponential model in comparison to the null model replicates to a new dataset, we would interpret

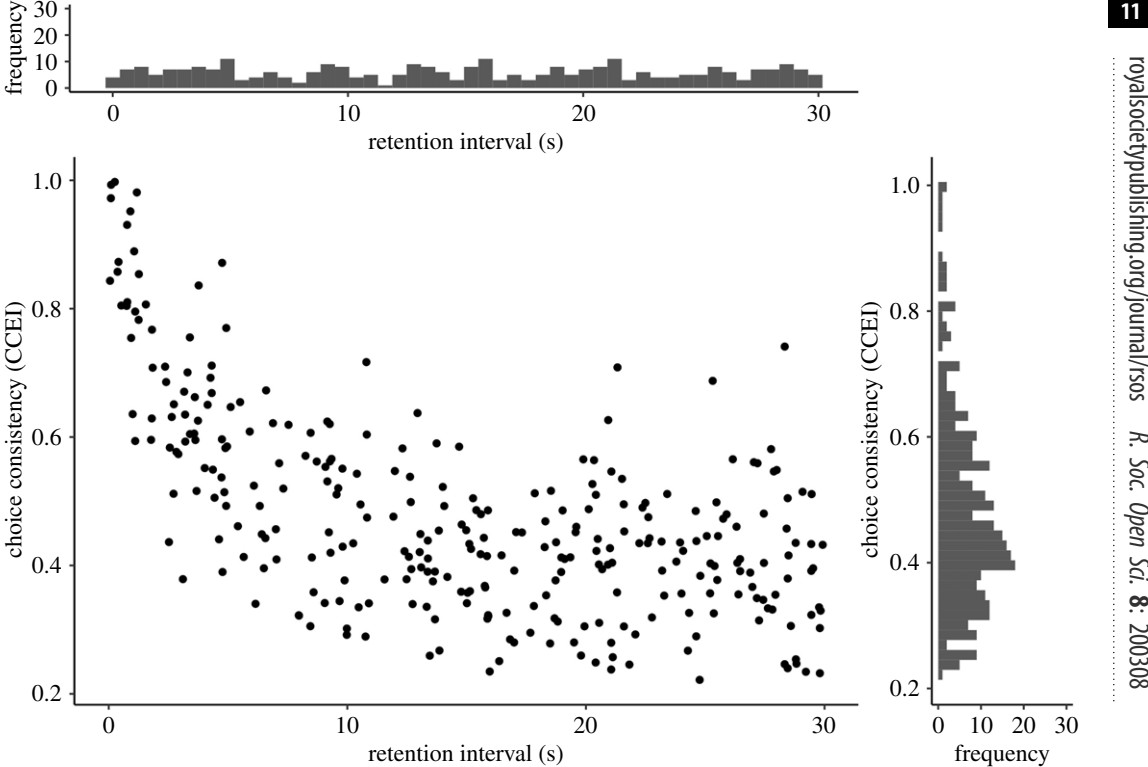

**Figure 5.** Scatterplot of RI and choice-consistency with histograms of marginal distributions. Note. Data were simulated for 300 virtual participants using the exponential model with parameters $a = 0.4$, $b = 0.9$, $\alpha = 0.3$, $\sigma = 0.1$. The marginal distribution of the RI is, trivially, uniform. Importantly, the marginal distribution of choice-consistency is a right-tailed Gaussian, meaning a tail for large consistency values.

**Table 1.** Summary of statistical interpretation criteria for each hypothesis.

| hypothesis | $BF \geq 10$ | $BF \leq 0.1$ | $0.1 < BF < 10$ |
|---|---|---|---|
| H1: inverse relationship of RI and choice-consistency | strong support for inverse relationship | strong support against inverse relationship | inconclusive, larger N required |
| H2: exponential model is supported more strongly by the data than null model | strong support for exponential model | strong support for null model | inconclusive, larger N required |
| H3: the finding of H2 replicates to a new dataset | strong support for replication to a new dataset | strong support against a replication to a new dataset | inconclusive, larger N required |

this as further evidence for the validity of the exponential model. Should the replication Bayes factor favour the null model, this would question the validity of an exponential model specifically, given positive evidence for H1. Again, our statistical tests would only collect relative evidence for one model over another. Careful inspection of the visualizations of the model predictions would be required (figures 5 and 6).

Should we find conflicting evidence for H2 and H3, we would use the Bayes factor for the complete dataset ($BF_{10}(d_{\text{test}}, d_{\text{train}})$) to guide our interpretation. The Bayesian model comparison using the complete dataset quantifies the evidence for or against each model in light of all data. We would use the same interpretation framework as before, which means that we consider a Bayes factor of $BF \geq 10$ as conclusive evidence.

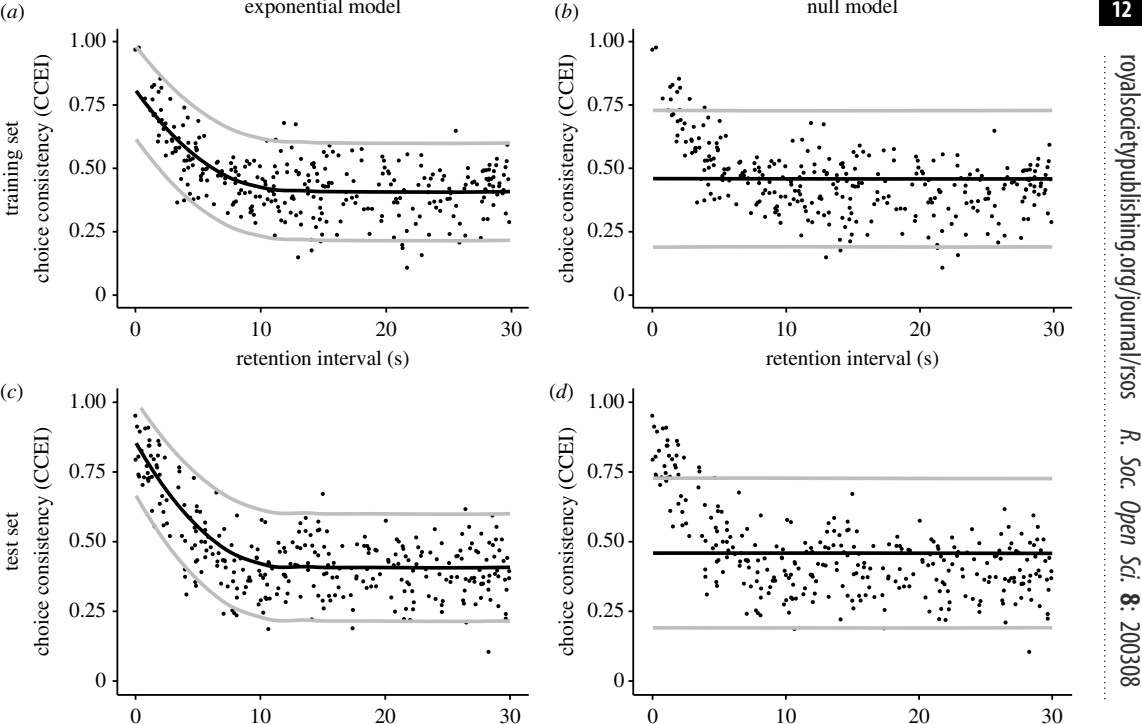

**Figure 6.** Scatterplots of RI and choice-consistency overlaid with posterior predictions. Note. The black line shows the median predictions, and the grey lines show the 95% highest density intervals. (*a,b*) The training set; (*c,d*) the test set. (*a,c*) Posterior predictions of the exponential model; (*b,d*) posterior predictions of the null model. Training and test sets were simulated for 300 virtual participants using the exponential model with parameters $a = 0.4$, $b = 0.9$, $\alpha = 0.3$, $\sigma = 0.1$. While the exponential model predicts the pattern of the data with relatively little uncertainty, the null model makes very vague predictions with possible values covering almost half of the variable space. Further, the null model does not predict the trend of the data for small RIs. Relative to the training set performance of each model, the exponential model also generalizes slightly better to the test set.

## 2.8. Data collection plan/power analysis

### 2.8.1. Inferential power

Our data collection plan is based on a Bayesian stopping rule: we collected data until we reached a Bayes factor of $BF \geq 10 \lor BF \leq 0.1$ or a maximum feasible sample size of $N = 500$.

### 2.8.2. Sensitivity of choice-consistency test

In order to make meaningful statements about the influence of memory processes, it is not only necessary to experimentally manipulate these memory processes with a sufficient effect size but also to measure choice-consistency with sufficiently sensitive measure. The sensitivity of our behavioural task to detect violations of choice-consistency can be approximated using a simulation study [49]. We simulated a dataset of 1000 virtual participants that made uniform random choices from 20 choice sets constructed as specified for our experiment (see Procedure). Results showed that 99% of the virtual participants violated choice-consistency at least once with a median CCEI of 0.389 (figure 7).

## 2.9. Specification of reality checks

First, to ensure that our RI manipulation is effective, we tried to replicate the effect of the RI on absolute reconstruction error of exemplars from memory that we found in our pilot experiment (see pilot experiment) in our control task. Specifically, we wanted to find strong evidence (Bayes factor of at least $BF \geq 10$) favouring a one-way ANOVA style model including the four-step RI factor over a null model. Inference was based on the replication Bayes factor fully using the evidence from our pilot experiment with $BF_{10}(d_{orig}) = 1000$ [47].

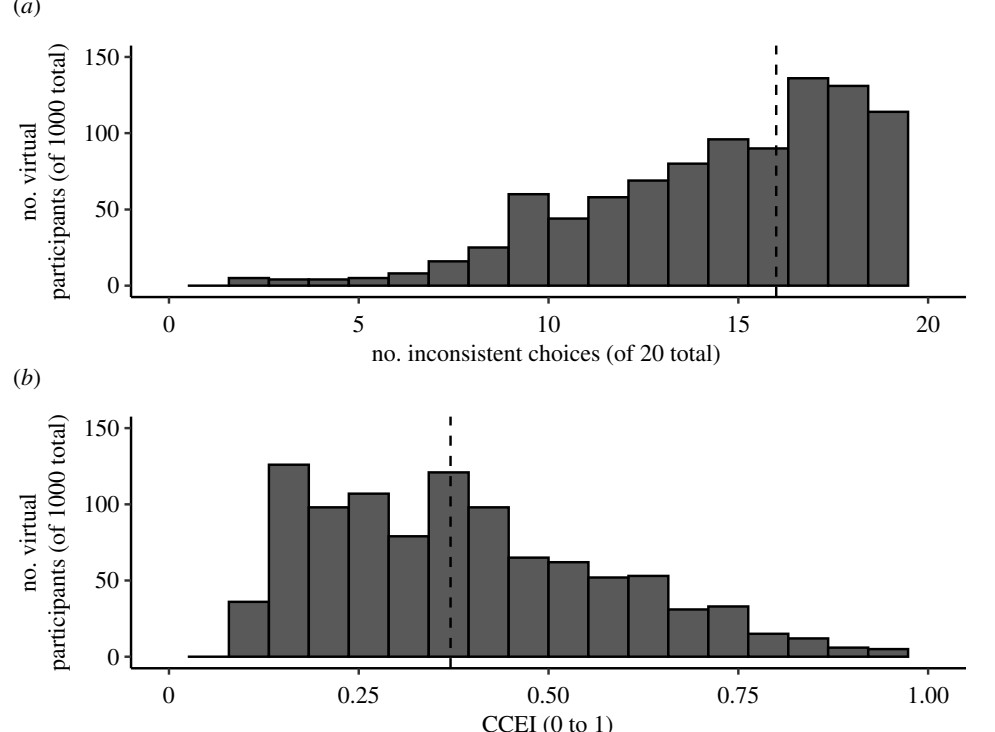

**Figure 7.** Histograms of choice-consistency of simulated random behaviour. Note. We simulated a dataset of 1000 virtual participants that made uniform random choices from 20 choice sets constructed as specified for our experiment (see Procedure). (*a*) The distribution of the number of inconsistent choices; 99% of the virtual participants committed at least one inconsistent choice. The median number if inconsistent choices (16) is indicated by the dashed vertical line. (*b*) The distribution of the CCEI. 99% of the virtual participants had a CCEI lower than 0.90. 97% of participants had a CCEI lower than 0.80. The median CCEI (0.37) is indicated by the dashed vertical line. Overall, our experimental task provides sufficient sensitivity to detect inconsistent choices. Note that of all 1000 virtual participants only a single one had a CCEI of 1. Importantly, this participant also did not violate the revealed preference axioms (0 inconsistent choices). We are, therefore, confident that our design also minimizes cost-efficient inconsistent choices which would undermine the sensitivity of the CCEI measure specifically [50].

We used the JAGS software [44] to analyse our Bayesian graphical models. To assess convergence, we used trace plots of the Markov chain Monte Carlo simulations and smoothed density plots of the parameter estimates [51].

Following Blaha [52], we think that visualization is an important reality check to see whether the data look like we expected and hypothesized. Therefore, we planned to create two main plots for visual qualitative checks of the data and models.

First, we created a scatter plot of RI and choice-consistency in the training set together with histograms of the marginal distributions. This allowed for a visual inspection of the relationship of both variables as well as the marginal distributions. We did not want to see choice-consistency increase as a function of RI, as such pattern is not covered by our model space. The marginal distribution of the RI should, trivially, be uniform (as generated by the experimental task). The marginal distribution of the choice-consistency should, ideally, be a right-tailed Gaussian, meaning a tail for large consistency values. Figure 5 shows such a plot for data simulated from the exponential model.

Second, we planned to create a plot that is overlaying scatter plots of RI and choice-consistency in the training set with the posterior predictive distributions of the exponential model and the null model (but see section Interpretative Plan and results for H1 as to why we did not proceed with our computational modelling). This would allow us to visually inspect how well the models can explain the data and, further, if there are any important qualitative differences between predictions and data. This is an important step to inform future modelling efforts and to identify systematic short comings of a model. Further, we would create the same plots for RI and choice-consistency in the test set overlaid with the out-of-sample posterior predictions of both models to qualitatively evaluate the generalizability of the models (but see section Interpretative Plan and results for H1 as to why we did not proceed with our computational modelling). Figure 6 shows such plots for data simulated from the exponential model.

### 2.9.1. Floor and ceiling effects

Should we find that choice-consistency is either near perfect or at very low levels across all RIs, this would indicate ceiling or floor effects respectively. While this is theoretically possible (e.g. in the case of an ineffective RI manipulation), we reduced the likelihood of finding such a pattern by using a continuous manipulation of the RI instead of a factorial design. Therefore, our design covered a wide range of RIs (interval of 30 s) instead of 2 to 3 RIs that a factorial design would cover. Still, it was not possible to entirely rule out the possibility of an ineffective RI manipulation on theoretical grounds only. Therefore, we conducted a pilot experiment to demonstrate the effectiveness of our manipulation using the control task from our main experiment (see below).

### 2.9.2. Further limitations

As one anonymous reviewer pointed out, our MAVC paradigm does not include a no-choice option. Intuitively, for some trials, it would be difficult for participants to make a similarity judgement. However, we decided not to include a no-choice option in our paradigm as one core assumption of revealed preference theory is that there is a well-defined preference structure [5], and this also holds for difficult decisions. Therefore, asking participants to make a choice for difficult decisions is part of a rigorous test of revealed preference choice-consistency. Still, practically, this could have introduced additional noise into the decision behaviour of participants. While the current registered report cannot entirely address this aspect of insufficiently defined preferences, future research should provide both theoretical and empirical accounts on the role of non-decisions for choice-consistency.

## 2.10. Pilot experiment

We conducted a pilot experiment to validate the effectiveness of our RI manipulation. Specifically, we wanted to demonstrate that our RI manipulation is sufficient to blur the memory representation of the exemplar. Furthermore, we explored the influence of presentation time of the exemplar on the memory representation strength.

### 2.10.1. Methods

#### 2.10.1.1. Procedure

The first three screens of each trial were equivalent to the procedure of the experiment for the here described registered report (figure 2). Each trial started with the presentation of a fixation cross. Participants were then presented with an exemplar cube with a certain orientation along the $X$- and $Y$-axis (see procedure of registered report) for either 1, 5, 10 or 30 s. Then, participants were presented with a mask of 10 randomly oriented cubes (figure 2) for a certain RI. Importantly, the RI in the pilot experiment was not fixed per participant. The RI lasted either 1, 5, 10 or 30 s. After the RI, participants were again presented with a single cube similar to the exemplar. The cube randomly matched the exemplar either regarding the $X$- or the $Y$-orientation, while the initial complementary orientation was chosen uniform randomly. Participants then had to turn the cube on the screen to match the exemplar regarding the complementary orientation using the arrow keys. Importantly, it was unknown to the participants whether they had to reconstruct the $X$- or $Y$-orientation both during the presentation time and the RI. Participants solved 10 for each factorial combination of the presentation times and RIs in random order.

#### 2.10.1.2. Sample characteristics and exclusion criteria

We included a total of 25 participants (21 women, 4 men; age: $M = 24$, range = 18–39) for our pilot experiment. The sample size was not determined *a priori*. Instead we used a Bayesian stopping rule, recruiting further participants until we reached a Bayes factor of at least $BF \geq 10$ for our hypothesis test. Importantly, the sample size is smaller than the minimal sample size we plan to recruit for the here described registered report. Participants were recruited from the same population we target for the here described registered report.

However, the study was conducted as an online experiment due to the ongoing COVID-19 crisis. It is intuitive that participants might be less attentive during online experiments than during laboratory-based experiments due to the uncontrolled way in which participants solve the task. Therefore, we assessed reaction times besides task performance and excluded single trials with reaction times

deviating more than three standard deviations from the grand mean. Note that this threshold amounted to about 30 s for a single trial. Hence, we are confident to not have excluded any meaningful data while considerably reducing measurement noise.

The study was approved by the local ethics board of Heinrich-Heine-University and was conducted in accordance with the Declaration of Helsinki. Participants were reimbursed by course credit.

### 2.10.1.3. Statistical analysis

We operationalized the memory representation strength of the exemplar as the absolute error with which its orientation could be reconstructed by the participants. We considered an orientation of 0 degrees and 360 degrees as equivalent. For example, if the orientation of the exemplar cube on the axis of interest is 90 degrees and the orientation of the reconstructed cube on that axis is 360, the absolute error is 90 and not 270. The absolute error can therefore range between 0 degrees and 180 degrees. The exact formula is given by

$$\text{Absolute Error} = \left| |Ori_{\text{Exemplar}} - Ori_{\text{Reconstructed}}| - 180 \right|.$$

We calculated a repeated measures Bayesian ANOVA in the 'BayesFactor' R package using the non-informative default priors [53]. We considered a Bayes factor equal to or larger than 10 regarding the main effect of the RI to be conclusive for or against our hypothesis. To verify the direction of the effect we considered the trend of the means of each factor level. Further, we exploratively inspected the evidence for or against a main effect of the presentation time and a possible interaction effect of both factors.

### 2.10.2. Results

We found that an ANOVA style model including both main effects and a random subject intercept but no interaction term to be the most likely model given the data. Specifically, this model was $BF_{10} = 895082$ ($\pm 0.94\%$) times more likely than the null model (including only the random subject intercept) given the data (figure 8).

#### 2.10.2.1. Retention interval

To quantify the evidence for an effect of the RI factor, we compared the evidence of the most likely model with the evidence for a model including only the main effect of the presentation time and the random subject intercept. We found that there was $BF_{10} = 1000 \pm 1.13\%$ times more evidence for the inclusion of the RI factor. We interpret this as definitive evidence. Inspection of the trend of means reveals that there is a positive relationship of RI and absolute error of reconstruction (figure 8).

#### 2.10.2.2. Presentation time

To quantify the evidence for an effect of the presentation time factor, we compared the evidence for the most likely model with the evidence for a model including only the main effect of the RI and the random subject intercept. We found that there was $BF_{10} = 1005 \pm (1.15\%)$ times more evidence for the inclusion of the presentation time factor. We interpret this as definitive evidence. Inspection of the trend of means reveals that there is a negative relationship of presentation time and absolute error of reconstruction (figure 8). Note, however, that is was an explorative analysis.

#### 2.10.2.3. Interaction retention interval × presentation time

To quantify the evidence against an interaction effect of both factors, we compared the evidence of the most likely model with the evidence for a model including both main effects, the random subject intercept and an interaction term. We found that there was $BF_{01} = 44446$ ($\pm 2.11\%$) times more evidence for the exclusion of the interaction term. We interpret this as definitive evidence. Note, however, that is was an explorative analysis.

### 2.10.3. Discussion

We conducted a pilot experiment to validate the effectiveness of our RI manipulation. We showed that the precision of the orientation reconstruction of an exemplar from memory decreases with retention time over an interval of 30 s. Therefore, we are confident that the planned RI manipulation of the here described registered report is effective to weaken the memory representation strength of an exemplar. Further, we explored the influence of different presentation times on orientation reconstruction precision. We found that precision increases with presentation time. In the context of our registered

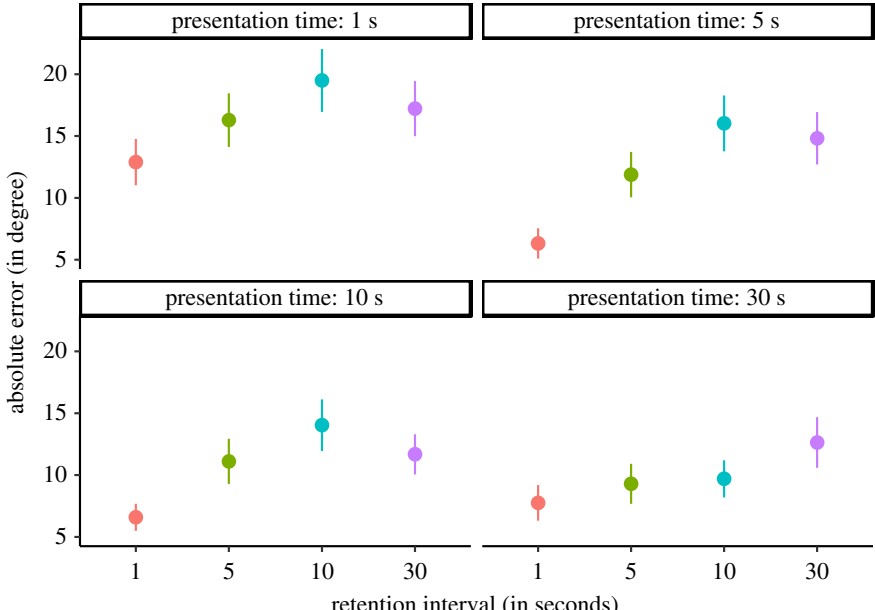

**Figure 8.** Results of a pilot experiment. Note. The figure shows the point range (mean and standard error of the mean) for each cell of the two-factorial design. The absolute error in the reconstruction of the exemplar cube orientation is positively related to the RI and negatively related to the presentation time. There is definitive evidence against an interaction of both factors. Note that the RI is not presented to scale.

report, it is important that the memory representation strength freely varies among different RIs for a given presentation time. Descriptively, the variance of the absolute error in reconstruction is highest for a presentation time of 1 s. However, also the mean absolute error is highest for a presentation time of 1 s. A presentation time of 5 s represents a compromise with the second highest variance and second highest mean of the absolute error in reconstruction.

# 3. Results

## 3.1. Sample characteristics

For our main experiment, we included 77 participants (56 women, 21 men; age: $M = 22$, range = 18–40; education: 72 completed high school, 5 completed a university degree) according to our inclusion criteria until reaching the preregistered stopping rule of our analysis plan.

## 3.2. Preregistered analyses

As outlined above, our statistical analyses use a Bayesian framework of inference, specifically the Bayes factor approach to model comparison. Bayes factors express the *relative* degree of evidence for one model over another that is the ratio of probabilities of observing the data under each model [54].

### 3.2.1. Reality check: effect of retention interval on memory representation

To quantify the evidence for an effect of the RI factor on memory representation strength, we compared a one-way ANOVA style model including the 4-step RI factor to a null model. We found conclusive evidence for the RI model for the new and the full dataset (including our pilot data), as well as for the successful replication $(BF_{10}(d_{new}) = 32.894 \pm 0.01\%$, $BF_{10}(d_{full}) = 46717770 \pm 0.01\%$, $BF_{Replication} = 44623.3$; figure 9). Hence, we can conclude that our RI manipulation was effective.

### 3.2.2. H1: test for an inverse relationship of retention interval and choice-consistency

Next, we tested in the training set whether choice-consistency as operationalized by the CCEI decreased with an increasing RI. Results showed conclusive evidence against our hypothesis $(BF_{10} = 0.047$;

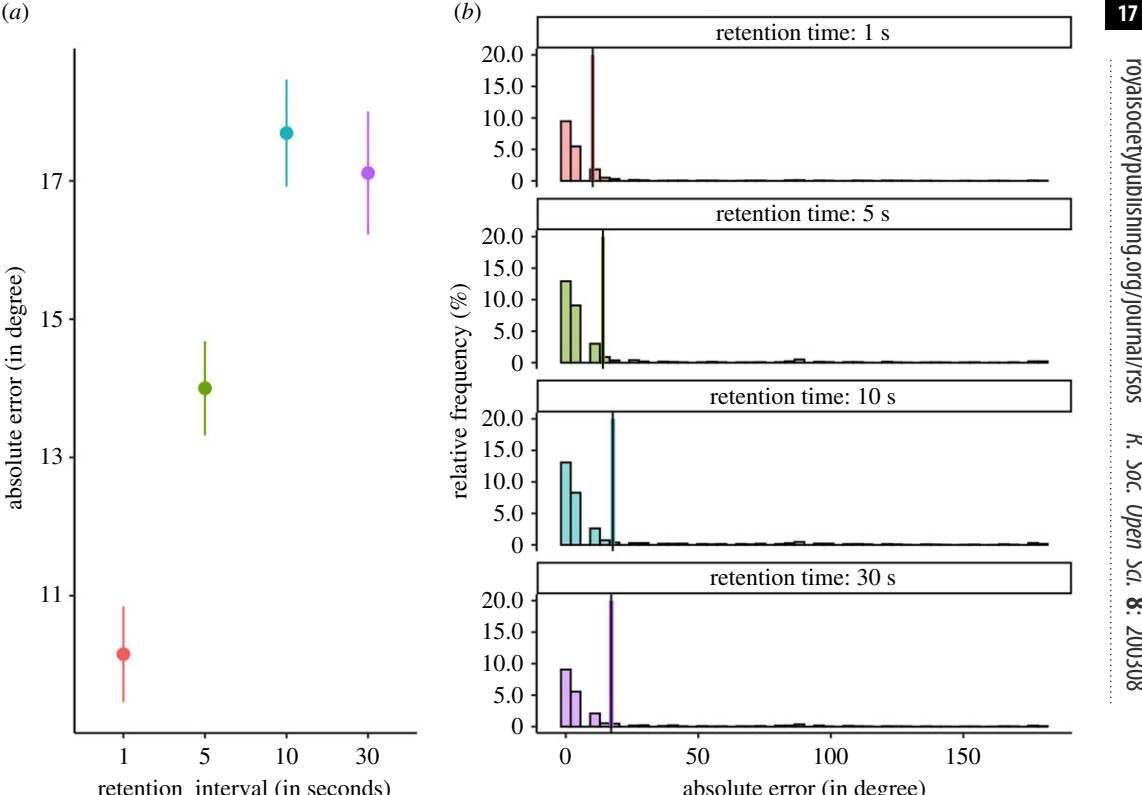

**Figure 9.** Manipulation check of the RI manipulation. Note. (*a*) The point range (mean and standard error of the mean) of the absolute reconstruction error for each RI level in the full dataset (pilot and preregistered experiment, disregarding encoding time for the former). The absolute error in the reconstruction of the exemplar cube orientation is positively related to the RI (indicated by the evidence for the pilot data, the new data and the replication). Note that the RI is not presented to scale. (*b*) The histogram of the absolute reconstruction error for each RI level in the full dataset (pilot and preregistered experiment, disregarding encoding time for the former). Vertical bars indicate the mean of the data, coloured tiles indicate the standard error of the mean. It is evident that the absolute error distribution has a strong positive skew.

figure 10). The result held for other specifications of choice-consistency, namely the Houtman–Maks index, and approximations of the money pump index and minimum cost index (all $BF_{10} < 0.1$).

### 3.2.3. H2 and H3: Bayes factor model comparison of exponential and null model

Following our interpretation plan (table 1), we did not proceed to test H2 and H3, given our negative result for H1.

### 3.2.4. Floor and ceiling effects

As apparent in figure 10, the CCEI of our participants in the training data was overall surprisingly low (median = 0.299, SD = 0.229). In order to control for a potential floor effect, we used a Bayesian Mann–Whitney U test to control whether our participants were more consistent than an equal-sized subsample of our random simulated data (figure 11*a*). Results showed strong evidence against this, indicating a potential floor effect in our data ($BF_{10} = 0.094$).

## 3.3. Exploratory analysis

### 3.3.1. Reliability analysis

In an attempt to further understand the quality of our data beyond our preregistered quality controls, we conducted a descriptive test–retest reliability analysis of the CCEI for the training and test set. As we reported recently elsewhere [55], there are concerns regarding the measurement reliability of the CCEI, which is especially problematic for correlational designs such as the one of the current study [56].

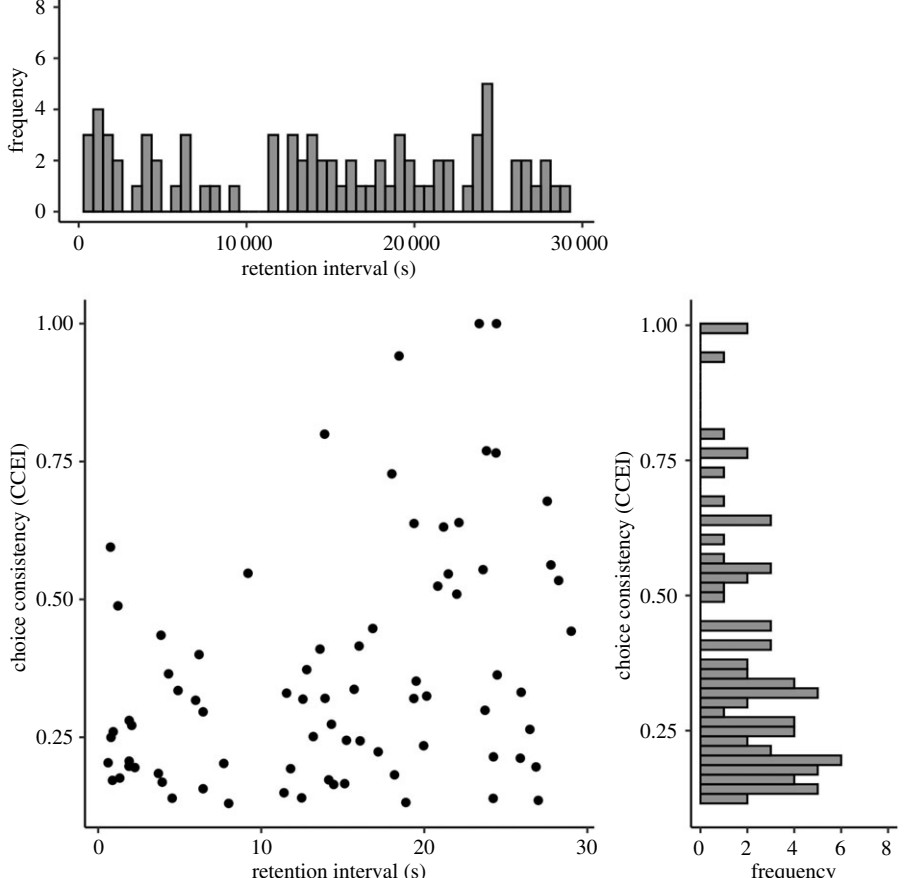

**Figure 10.** Scatterplot of the empirical RI and choice-consistency with histograms of marginal distributions. Note. Trivially, the marginal distribution of the RI was uniform. Further, as expected, the marginal distribution of choice-consistency was a right-tailed Gaussian, meaning a tail for large consistency values. However, contrary to our hypothesis, the bivariate distribution plot revealed no negative relationship of CCEI and retention time.

Specifically, tasks designed to show robust between-group effects and, thus, low between-subject variability in the outcome measure are at risk for showing low test–retest reliability. Another risk factor specific to the CCEI is that the measure is dependent only on the magnitude of the most severe violation (see Preprocessing) and, thus, vulnerable to outliers. Our results indicated essentially no reliability of the CCEI between training and test set in the current study ($r = -0.033$). This was not driven by the difference in RIs of both measurements, or by low between-subject variability of the measure (figure 11a,b). A similar result showed for the money pump index ($r = 0.023$). However, interestingly, the Houtman–Maks index and the minimum cost index showed a much higher (albeit still poor) test–retest reliability (HMI: $r = 0.303$, MCI: $r = 0.353$; figure 11c–e), which might be attributed to less vulnerability to outliers.

### 3.3.2. Task difficulty

Given the results reported above and oral feedback from our participants, we formed the *post hoc* hypothesis that the generally low choice-consistency might be driven by too high a difficulty of discriminating the different $X$- and $Y$-orientations of the choice objects. To further explore this notion, we compared the mean absolute error in the reconstruction task, as an upper limit to the discriminatory performance, to the mean increment difference of orientation along the $X$- or $Y$-axis in the choice set, using bootstrapping. Results showed that the mean increment difference was generally lower than mean reconstruction error, tentatively suggesting a high difficulty of discriminating between the choice objects (figure 12).

## 4. Discussion

In this registered report, we set out to experimentally test the influence of memory retrieval of exemplars on choice-consistency in a novel visual choice paradigm. After a short RI, participants had to select one

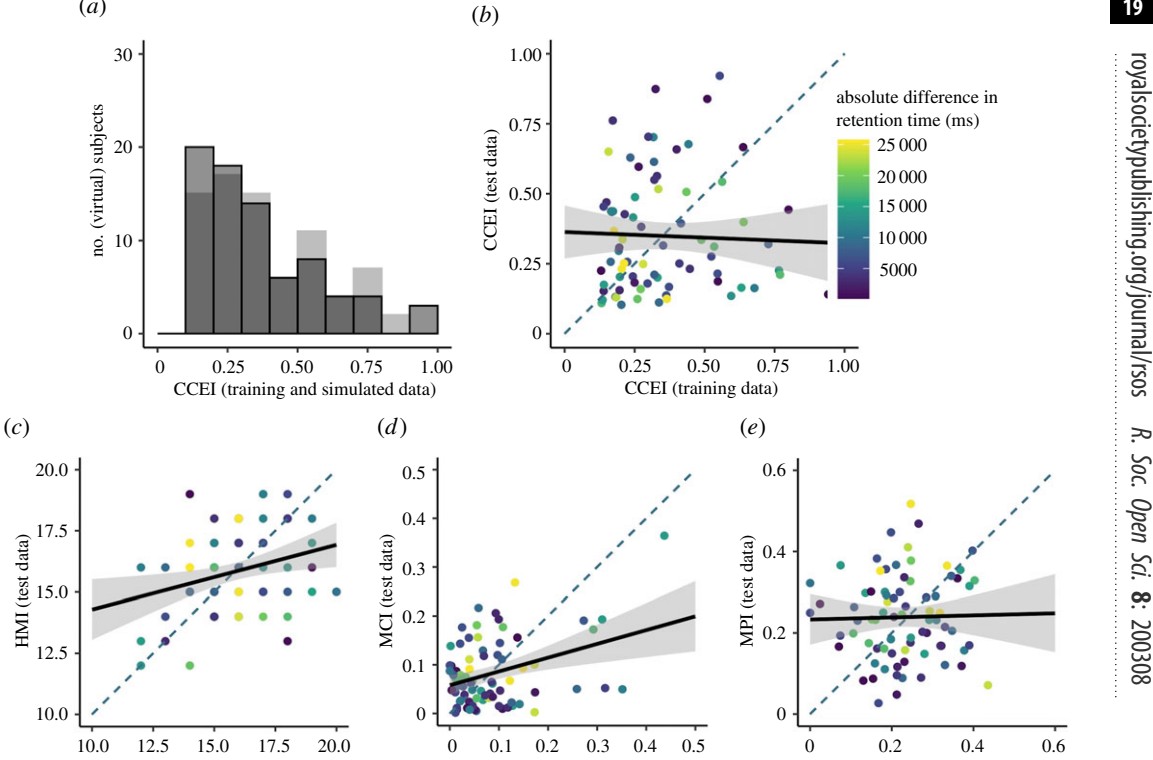

MCI and MPI approximated considering only direct GARP violations

**Figure 11.** Empirical choice-consistency and reliablity. Note. (*a*) The empirical distribution of the CCEI compared to an equally sized subset of simulated random behaviour. Choice-consistency, overall, was surprisingly low and not higher than for simulated random behaviour. (*b*) The test–retest reliability of the CCEI for training and test data, which was almost zero. Importantly, this was not driven by the absolute difference in retention time between both measurements. (*c,d,e*) Similar patterns for three other consistency indices.

out of a choice set of five three-dimensional cubes that have the subjectively most similar orientation along the *X*- and *Y*-axis to the exemplar. The choice set stimuli represented a variable trade-off of similarity to the exemplar regarding the two attributes *X*- and *Y*-orientation. We manipulated memory retrieval by varying the duration of the RI between exemplar presentation and choice.

Using a reconstruction task as a manipulation check of our RI intervention, we could show and replicate the pattern of decreasing memory accuracy with increasing retention time in a pilot experiment and in our preregistered study, which confirmed the effectiveness and reliability of our manipulation.

Given this, we found strong evidence against our first hypothesis that choice-consistency, as operationalized by the CCEI, decreases with increasing retention time. Further, this result held for robustness checks using three similar choice-consistency indices. Given our preregistered interpretation plan we did not proceed to test our more specific, model-based hypotheses.

## 4.1. Limitations

However, our preregistered quality controls revealed an overall surprisingly low choice-consistency of our participants even for short RIs that proved to be non-discernable from that of simulated random behaviour. In addition, an exploratory analysis showed essentially no test–retest reliability of the CCEI between the training and the test set. This was not driven by retention time differences between the two measurements. Taken together, this suggests the presence of a floor effect in our data and, thus, low data quality for conclusively evaluating our hypotheses.

Generally, the lack and low reliability of choice-consistency indicates that our participants did not consistently integrate deviations in the *X*- and *Y*-dimensions of the choice set stimuli to the exemplar, meaning there was no well-behaved integration function, and this was also the case for short retention times. As the performance in the reconstruction task was generally good (perfect reconstruction in about 42% of trials), it is unlikely that the low consistency level was driven by too long RIs.

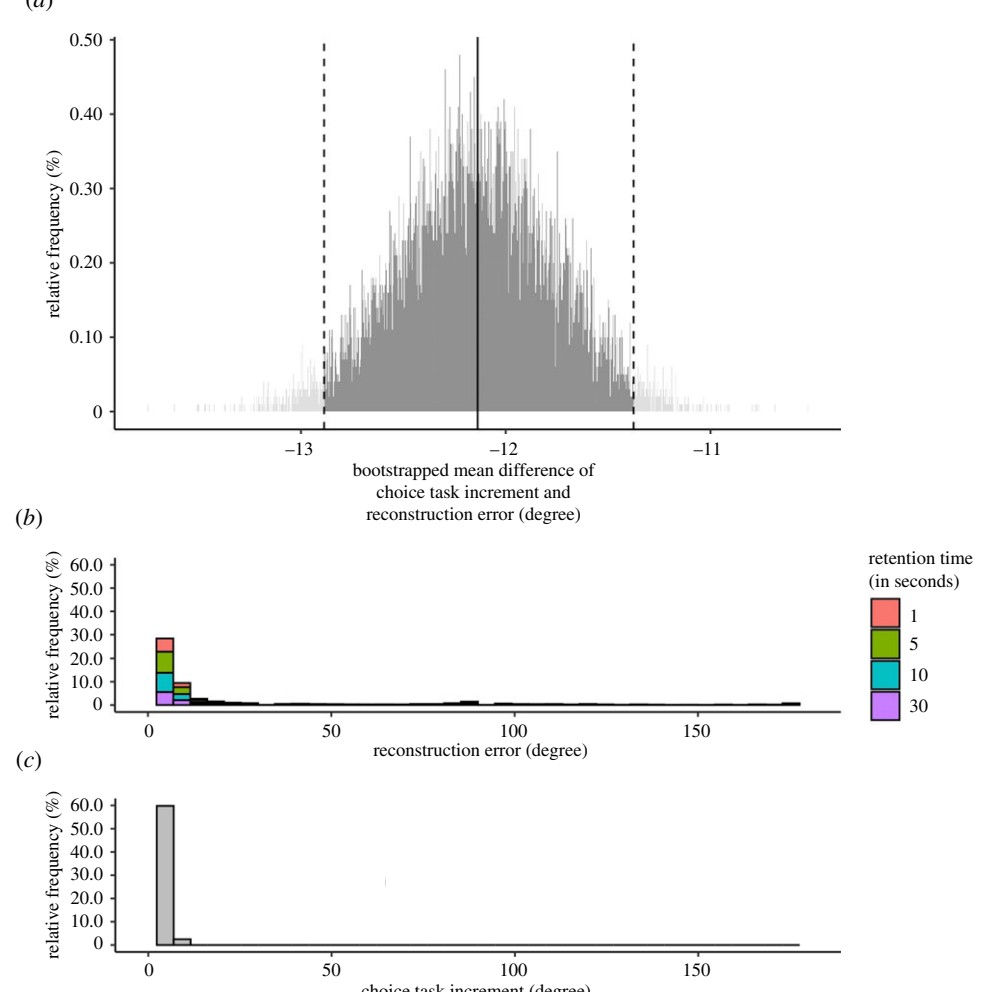

**Figure 12.** Exploration of task difficulty. Note. (a) The bootstrapped distribution of the mean differences between the empirical absolute reconstruction error and choice task increment difference in the choice set (retrieved from task parameters). Depicted is the histogram of the bootstrap samples distribution ($N = 10\,000$). The continuous vertical line indicates the mean of the statistic of interest; the two dashed vertical lines indicate the 95% confidence interval. Results showed that the mean increment difference was generally lower than mean reconstruction error, indicating a high difficulty of discriminating between the choice objects.

Another explanation for the overall low choice-consistency could be that the discrimination between the different $X$- and $Y$-orientations of the choice objects was too difficult. This was also, anecdotally, suggested in oral feedback of our participants during the data collection. To further explore this notion, we compared the mean absolute error in the reconstruction task, as an upper limit to the discriminatory performance, to the mean increment difference of orientation along the $X$- or $Y$-axis in the choice set, recovered from the task parameters, using bootstrapping. Results showed that the mean increment difference was generally lower than mean reconstruction error, tentatively suggesting a high difficulty of discriminating between the choice objects. As all choice-consistency indices quantify performance only relative to the increment orientation differences of choice objects, numerically low choice-consistency levels can correspond to only small inconsistencies in degree orientation.

## 4.2. Future research

Future studies investigating visual choice-consistency should, therefore, establish a sufficient level of choice-consistency at baseline. This could be achieved, for example, by pilot testing to adjust, in a group-wise fashion, the increment difference of orientation along the $X$- or $Y$-axis in the choice, or on an individual level by using an adaptive staircase procedure.

Another important consideration for the design of future studies is the low reliability of choice-consistency in the present study, but also, generally, in other task domains [55]. While our correlational design had important benefits for covering a sufficient retention time span and providing rich data for parametric model fitting, factorial designs are more robust to finding effects in low-reliability behavioural tasks [56].

# 5. Conclusion

In this registered report, we set out to experimentally test the influence of memory retrieval of exemplars on choice-consistency in a novel visual choice paradigm. Due to unforeseen methodological pitfalls, our data are inconclusive to the preregistered hypotheses. However, our preregistered quality controls and additional exploratory analyses offer important insights for the design of future studies.

Ethics. All participants gave informed written consent. The study was approved by the local institutional review board of Heinrich-Heine-University and was conducted in accordance with the Declaration of Helsinki. Participants were reimbursed by course credit.

Data accessibility. Raw and processed data, the approved stage 1 manuscript, as well as analysis code for the stage 1 and stage 2 manuscript results and figures are available online: https://osf.io/2vx36/.

Authors' contributions. F.J.N.: involved in conceptualization, methodology, software, formal analysis, investigation, data curation, writing the original draft, visualization and project administration. T.K.: involved in resources, writing—review & editing, supervision, project administration and funding acquisition.

Competing interests. We declare we have no competing interests.

Funding. The research was supported by a grant from the Deutsche Forschungsgemeinschaft (Bonn, Germany) to T.K. (grant no. DFG-KA 2675/4-3).

Acknowledgements. We thank Alina Keßler, Paul Kramer and Marie Ludes for their support on the pilot data collection. We thank Quentin Gronau and three anonymous reviewers for their helpful criticism which led to a greatly improved version of stage 1 manuscript. Further, we thank Ana Hernandez for her support on the data collection of the main experiment.

# Appendix. English translation of instructions for MAVC task

Dear participant,
 In the following task, you will be presented with a number of independent decision problems that share a common format. Each decision problem starts with the presentation of a colourful three-dimensional cube. The cube will be presented for 5 s. Each side of the cube is identified by a unique colour. Your task is to memorize the orientation of the cube as good as possible. After the presentation time of 5 s has passed, you will be presented with a visual mask of 10 similar cubes for up to 30 s. These cubes are irrelevant for the decision problem and you should not try to memorize their orientation. Finally, you will be presented with five more cubes in different orientations which will be presented to you in a circle. Your task is to select the one of those five cubes which has the most similar orientation to the cube which you have been presented with at the beginning of the decision problem. Before each decision problem you will be shortly presented with a fixation cross.

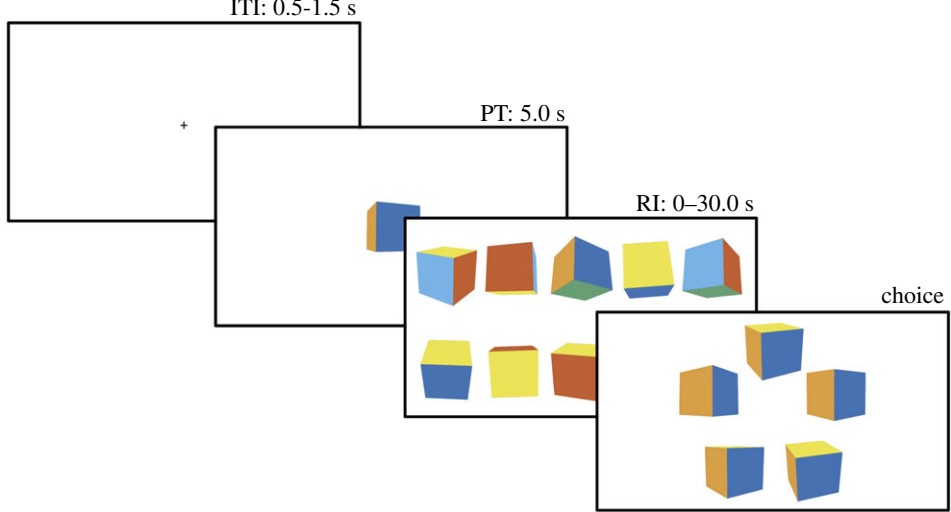

Carefully evaluate all five cubes and try to mentally rotate them until they match the memorized cube. Then select the cube which had to be mentally rotated the least. You have to decide for each decision problem. If you are unsure about your answer, follow your intuition. There are no wrong or correct answers. Before the task begins, please take a moment to familiarize yourself with the colourful cube by inspecting the following animation. All cubes presented in the task will be exact copies of that cube but in different orientations.

[ANIMATION HERE]

Thank you for familiarizing yourself with the colourful cube. You will now be presented with 10 practice decision problems. For these practice problems, the choice options will be presented 1 s after the cube that you have to memorize. Your answers for these practice decisions will not be recorded. Take your time to familiarize yourself with the task.

You have successfully completed the practice decision problems. Do you have any questions or is there anything unclear about the task at hand? Then please raise your hand and consult with the experimenter.

If you have no further questions, then you can proceed now with the first test block. The test block consists of 20 decision problems. For all 20 decisions, you will be assigned a RI of up to 30 s after the presentation of the initial cube during which you will see the irrelevant visual mask.

You have successfully completed the first test block. Take a moment to stretch your legs before continuing.

The next test block again consists of 20 decision problems. For all 20 decisions, you will be assigned a different RI of up to 30 s after the presentation of the initial cube during which you will see the irrelevant visual mask.

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
