## [Peer Review File · Royal Society Open Science]

Review History

RSOS-200308.R0 (Original submission)

Review form: Reviewer 1

Do you have any ethical concerns with this paper?

No

Recommendation?

Accept with minor revision

Comments to the Author(s)

I like the topic of the report and the experimental design is well-crafted. I have a few comments however that I would like the authors to consider before their data collection:

- 1) Make the hypotheses more friendly to read and understand, and provide more discussion about the importance of each of the hypothesis and how results will contribute to the literature on choice consistency.
- 2) Page 8 – what does “participants must be healthy” really mean? Be more specific.

3) My understanding is that the “trials” will consist of 5 rectangles and the subject will be asked to select one that looks most similar to the exemplar. My question is shouldn't there be a “no-choice” option in each trial, especially since there can be trials where the choice set stimuli are quite different from the target stimuli? In many discrete choice experiments, researchers now tend to include a “no-choice or none of the above” option in each of the choice sets given to subjects.

4) I wonder if results also be contingent upon the time that each exemplar is presented to subjects. In your present design, you plan to provide 5 seconds for this. Would results change if you gave them 10 seconds, for example?

5) On top of page 11, it was mentioned that participants will be debriefed and reimbursed after they complete the second test block. This is vague and needs to be clarified more. For example, what would the debriefing entail? How are subjects being incentivized to put cognitive effort in their choices? Will one of the trials be randomly chosen as the binding trial?

Review form: Reviewer 2

Do you have any ethical concerns with this paper?

No

Recommendation?

Major revision

Comments to the Author(s)

Summary: Nitsch and Kalenschar aim to investigate the role of memory representations for choice consistency. As memory representations are typically unknown, they suggest that the underlying process can be investigated using a multi-attribute visual choice (MAVC) paradigm. In every trial of their version of the paradigm, participants will first see an exemplar stimulus. Stimuli will be bars differing in color and orientation. After a retention interval, participants will be confronted with five different bars of which participants have to select one. None of the stimuli will perfectly match with the exemplar but they will all constitute a trade-off between the two feature dimensions. The authors will manipulate the length of the retention interval and expect that this duration and choice consistency with regard to the two dimensions will be inversely related. That is, they expect that choice consistency will be high for short duration intervals and will decrease exponentially with increasing retention interval duration.

Evaluation: Overall, the manuscript investigates an interesting question and seems to be a promising and fruitful approach. However, I have a few recommendations regarding the clarity of the manuscript or questions that arise due to unclear or missing information. Moreover, I have some concerns regarding the construction of stimuli and memory manipulation that I would recommend to address before collecting data. If the authors address these concerns, then the planned work will be scientifically strong and a meaningful contribution to the literature. I am looking forward to read the authors' reply and to see how the results turn out.

--- MEMORY INTERVAL & STIMULI ---

- Efficiency of decay for a single item: I doubt that information will necessarily decay in case of a single item with two informative features. According to the interpretative plan the absence of an inverse relationship between choice consistency and retention interval would “question the role of memory representation in [MAVC].” However, an alternative option is that this relationship exists but that the manipulation of retention interval was not successful. To overcome this, I would recommend to do a manipulation check. Here are some possible options that came to my mind:

- 1.) Introduce a manipulation check during the experiment. For example, the authors might introduce a few memory trials at the end of every participant's session where they ask them to adjust the orientation and color of a previously seen exemplar. The time between showing the exemplar and the response screen should correspond to the two retention intervals tested. If the manipulation was successful, then precision should decay with the retention interval (i.e. larger orientation and hue errors). This procedure would come at a risk: If it turns out that the manipulation was not successful, then no conclusions could be drawn about the relationship between memory representation and MAVC.
- 2.) Pilot the success of the memory manipulation before recording the actual experiment. This pre-test could be done with a small additional sample that do a few trials each with e.g. 0, 15 and 30 s retention interval. When asking for color and orientation it would be important to vary the response order (due to response times adding to the retention interval) or to let participants judge/adjust both dimension simultaneously. Such a pilot experiment might turn out to be the safer option as it might prevent recording up to 500 participants with an unsuccessful manipulation. If the authors decide for a pilot experiment, and the manipulation turns out to be not successful, then the authors might want to bet their money on a different horse, e.g. ...
- 3.) ...interference instead of decay: Instead of varying the time between exemplar and choice, the authors might additionally (or instead) try to interfere with the memory representation, e.g. by showing a mask. The mask could consist of several colored lines in various orientations across the screen and the authors could vary the strength of the mask by varying its contrast. Other manipulations are of course possible. But this would only be necessary if the retention manipulation is not successful. Also in this case a manipulation check would be necessary.

- Orientation: cardinal axes. Related to my point above. People have a higher sensitivity to orientations along the cardinal axes compared to any other orientation. This would certainly also affect the memory representation. For example, a bar along the vertical could be remembered by remembering that it was vertical, whereas remembering a bar with an additional rotation of +10 deg would have to rely on some sort of visual memory. I would assume that the decay would only be successful (if at all) for orientations different than the cardinal axes. The authors might want to consider this when creating the exemplars and choice stimuli.

- Color: The manuscript is lacking some important information about how the color stimuli will be manipulated/created. Which color space do the authors use? Along which dimension will colors be manipulated? Saturation? From what I read and see in the figures, I would assume that the authors use rgb values and vary one of the three dimensions only? The following website gives an overview specifically related to PsychoPy and might be helpful in describing how stimuli are manipulated: <https://www.psychopy.org/general/colours.html> I was just thinking that it might be helpful to use DKL color space (monitor needs to be calibrated for this). Because in DKL space, color hues can be arranged in a color wheel and thus circularly. Just like the exemplars orientation. This would give the possibility to define color changes as everything within 0 deg or 90 deg along the color wheel. This would result in a nice consistency between the two feature dimensions.

- Other dimensions to choose stimuli: The authors would have to make sure that stimuli can only be chosen based on their orientation and color. From what it seems (Fig 2) participants could also choose stimuli based on their location, e.g. some choice stimuli are closer to the initial exemplar position than others. One possible solution to this would be to organize objects in a circular arrangement around the exemplar location with equidistant locations and randomly rotate locations every trial minimize the potential implicit and explicit location strategies. Moreover, in Fig 2 it appears as if the different orientations are associated with a different aspect ratio of the rectangles. I assume that this is only due to the compressed figure?

--- CLARITY OF THE MANUSCRIPT ---

- Abstract: “we will manipulate memory retrieval by changes of the retention interval”. It was not clear upon first reading whether the duration of the interval is changed or something else is happening in the interval.
- p.3: From the example at the beginning of the introduction it does not get clear why choice-consistency is considered a hallmark of rational value-based decision making as stated in the abstract and why it is worth investigating.
- P. 4, line 33-43 (Participants...research design): This part was unclear to me when reading it for the first time.
- P. 4, line 47: Please introduce multi-attribute visual choice shortly when the term is used for the first time.
- P. 5, line 37: it would be helpful if the authors could reiterate the two processes they are referring to.
- P. 6, line 39. What is the authors’ framework? Fig. 1? Please specify
- P. 6, last paragraph. The hypothesis is clear, but I would recommend to nevertheless add one sentence like “That is, we expect a high choice consistency when... and a low choice consistency...” to further increase clarity
- P. 7., line 20: please briefly introduce the term out-of-sample data when using it for the first time.
- P. 8: I expected a paragraph about the setup and stimuli? Which monitor is used? How far away are subjects seated from the monitor? Given this distance, how big in degree visual angle do objects appear on the screen? Which characteristics do stimuli have? How are the stimuli controlled (PsychoPy)? And so on. I think adding this information would help a lot to understand the experiment.
- P. 9, lines 10-20: Related to the point above. It is hard to understand the stimuli and experimental procedure when it is only introduced as valued-based decision paradigm. It might have been clearer when stimuli and their characteristics were first introduced with the terms that would be normally used to describe a color-orientation memory experiment. I would recommend doing this first and then bridge the gap to how this can understood as a value-based decision paradigm.
- What is the task of participants? Please specify
- P. 10-11: Definitions 1-3: It would be helpful for the reader to give examples.
- Fig. 3: Having multiple lines would have helped me to better understand the experiment, the role of the budget and the resulting possible choice stimuli. Here, one choice stimulus is identical with regard to color. What about additional examples where it is identical with neither or with orientation? Or for different budgets? If this creates too much clutter, new panel(s) can be added. No changes are necessary if everything is sufficiently clear in the main text.

--- MINOR POINTS ---

- P. 4, l. 28: “older adults” is very unspecific. Please specify. E.g. (>60)
- P. 5, line 47-49: E Zarahn versus Eric Zarahn in in-text citation: I assume that the reference management system thinks these are two different authors and therefore highlights their first name?
- P. 8, line 28: Some guidelines including APA recommend to avoid one sentence paragraphs.
- P.9, line 5: Why does m need an index? When reading it first I understood it is always 100 and can thus be treated as a constant?
- Fig.1: Is there a reason for having two distinct choice nodes? Shouldn’t this be the same behavior?
- Fig.2: Why does it need the square borders around the stimuli?

- Fig. 2: I would recommend to arrange these screenshots as a time line that states what is shown at which time during an individual trial.
- Fig.3: Please try to be more specific: What does relative mean?

Review form: Reviewer 3

Do you have any ethical concerns with this paper?

No

Recommendation?

Accept with minor revision

Comments to the Author(s)

The authors propose to use multi-attribute visual choice as a paradigm for assessing the influence of memory retrieval of goals on choice consistency. Specifically, they aim to experimentally test the influence of memory retrieval of exemplars on choice consistency in a visual choice paradigm.

As a disclaimer, I am not an expert on the content aspect of the paper and I have been invited to focus on the proposed Bayesian analyses which is the reason I will only comment on those.

Comments.

Overall, the proposed Bayesian analyses appear sound to me. However, I am not sure that the strategy of computing Bayes factors for only a training set and assessing out-of-sample predictive performance by performing a one-sided Bayesian t-test on the mean squared prediction error for the test set is ideal. I do not necessarily want to urge the authors to not conduct these analyses, however, I would like to ask them to consider the following additions:

- 1) I would like to see the Bayes factors that are computed based on the complete data set (i.e., test + training). These quantify the overall evidence in the data that will be collected and are of key interest.
- 2) An alternative approach to assessing out-of-sample predictive performance is to use the test set to update the priors for the parameters and then use these new priors (i.e., the posteriors based on the training set) to predict the test data using a Bayes factor approach. That is, the adequacy of these predictions can naturally be tested using Bayes factors (see Ly et al., 2019).

References.

Ly, A., Etz, A., Marsman, M., & Wagenmakers, E.-J. (2019). Replication Bayes factors from evidence updating. *Behavior research methods*, 51, 2498-2508.

Review form: Reviewer 4

Do you have any ethical concerns with this paper?

No

Recommendation?

Major revision

Comments to the Author(s)

See attached report (Appendix A).

Decision letter (RSOS-200308.R0)

15-Apr-2020

Dear Dr Nitsch,

The Editors assigned to your Stage 1 Registered Report ("Influence of memory processes on choice consistency") have now received comments from reviewers. We would like you to revise your paper in accordance with the referee and editors suggestions which can be found below (not including confidential reports to the Editor). Please note this decision does not guarantee eventual acceptance.

When submitting your revised manuscript, you must respond to the comments made by the referees and upload a file "Response to Referees" in "Section 2 - File Upload". Please use this to document how you have responded to the comments, and the adjustments you have made. In order to expedite the processing of the revised manuscript, please be as specific as possible in your response.

Kind regards,
Professor Chris Chambers
Royal Society Open Science
openscience@royalsociety.org

on behalf of Professor Chris Chambers (Registered Reports Editor, Royal Society Open Science)
openscience@royalsociety.org

Associate Editor Comments to Author (Professor Chris Chambers):

Associate Editor: 1

Comments to the Author:

Four expert reviewers have now assessed the Stage 1 manuscript, including three field specialists (Reviewer 1, 2, 4) and one statistical specialist (Reviewer 3). All find merit in the proposal while also flagging a wide variety of issues to address in revision - including the clarity and rationale of the hypotheses, consideration of manipulation checks (noted by two reviewers), inclusion of appropriate piloting, increased methodological detail, justification of specific design

components, and the computation/reporting of Bayes factors. Although the concerns raised are substantial, they are within bounds for revision of a Stage 1 RR. The reviews are also extremely constructive and helpful. A Major Revision is therefore recommended.

Comments to Author:

Reviewer: 1

Comments to the Author(s)

I like the topic of the report and the experimental design is well-crafted. I have a few comments however that I would like the authors to consider before their data collection:

- 1) Make the hypotheses more friendly to read and understand, and provide more discussion about the importance of each of the hypothesis and how results will contribute to the literature on choice consistency.
- 2) Page 8 - what does "participants must be healthy" really mean? Be more specific.
- 3) My understanding is that the "trials" will consist of 5 rectangles and the subject will be asked to select one that looks most similar to the exemplar. My question is shouldn't there be a "no-choice" option in each trial, especially since there can be trials where the choice set stimuli are quite different from the target stimuli? In many discrete choice experiments, researchers now tend to include a "no-choice or none of the above" option in each of the choice sets given to subjects.
- 4) I wonder if results also be contingent upon the time that each exemplar is presented to subjects. In your present design, you plan to provide 5 seconds for this. Would results change if you gave them 10 seconds, for example?
- 5) On top of page 11, it was mentioned that participants will be debriefed and reimbursed after they complete the second test block. This is vague and needs to be clarified more. For example, what would the debriefing entail? How are subjects being incentivized to put cognitive effort in their choices? Will one of the trials be randomly chosen as the binding trial?

Reviewer: 2

Comments to the Author(s)

Summary: Nitsch and Kalenschar aim to investigate the role of memory representations for choice consistency. As memory representations are typically unknown, they suggest that the underlying process can be investigated using a multi-attribute visual choice (MAVC) paradigm. In every trial of their version of the paradigm, participants will first see an exemplar stimulus. Stimuli will be bars differing in color and orientation. After a retention interval, participants will be confronted with five different bars of which participants have to select one. None of the stimuli will perfectly match with the exemplar but they will all constitute a trade-off between the two feature dimensions. The authors will manipulate the length of the retention interval and expect that this duration and choice consistency with regard to the two dimensions will be inversely related. That is, they expect that choice consistency will be high for short duration intervals and will decrease exponentially with increasing retention interval duration.

Evaluation: Overall, the manuscript investigates an interesting question and seems to be a promising and fruitful approach. However, I have a few recommendations regarding the clarity of the manuscript or questions that arise due to unclear or missing information. Moreover, I have some concerns regarding the construction of stimuli and memory manipulation that I would recommend to address before collecting data. If the authors address these concerns, then the planned work will be scientifically strong and a meaningful contribution to the literature. I am looking forward to read the authors' reply and to see how the results turn out.

--- MEMORY INTERVAL & STIMULI ---

- Efficiency of decay for a single item: I doubt that information will necessarily decay in case of a single item with two informative features. According to the interpretative plan the absence of an inverse relationship between choice consistency and retention interval would “question the role of memory representation in [MAVC].” However, an alternative option is that this relationship exists but that the manipulation of retention interval was not successful. To overcome this, I would recommend to do a manipulation check. Here are some possible options that came to my mind:

- 1.) Introduce a manipulation check during the experiment. For example, the authors might introduce a few memory trials at the end of every participant’s session where they ask them to adjust the orientation and color of a previously seen exemplar. The time between showing the exemplar and the response screen should correspond to the two retention intervals tested. If the manipulation was successful, then precision should decay with the retention interval (i.e. larger orientation and hue errors). This procedure would come at a risk: If it turns out that the manipulation was not successful, then no conclusions could be drawn about the relationship between memory representation and MAVC.
- 2.) Pilot the success of the memory manipulation before recording the actual experiment. This pre-test could be done with a small additional sample that do a few trials each with e.g. 0, 15 and 30 s retention interval. When asking for color and orientation it would be important to vary the response order (due to response times adding to the retention interval) or to let participants judge/adjust both dimension simultaneously. Such a pilot experiment might turn out to be the safer option as it might prevent recording up to 500 participants with an unsuccessful manipulation. If the authors decide for a pilot experiment, and the manipulation turns out to be not successful, then the authors might want to bet their money on a different horse, e.g. ...
- 3.) ...interference instead of decay: Instead of varying the time between exemplar and choice, the authors might additionally (or instead) try to interfere with the memory representation, e.g. by showing a mask. The mask could consist of several colored lines in various orientations across the screen and the authors could vary the strength of the mask by varying its contrast. Other manipulations are of course possible. But this would only be necessary if the retention manipulation is not successful. Also in this case a manipulation check would be necessary.

- Orientation: cardinal axes. Related to my point above. People have a higher sensitivity to orientations along the cardinal axes compared to any other orientation. This would certainly also affect the memory representation. For example, a bar along the vertical could be remembered by remembering that it was vertical, whereas remembering a bar with an additional rotation of +10 deg would have to rely on some sort of visual memory. I would assume that the decay would only be successful (if at all) for orientations different than the cardinal axes. The authors might want to consider this when creating the exemplars and choice stimuli.

- Color: The manuscript is lacking some important information about how the color stimuli will be manipulated/created. Which color space do the authors use? Along which dimension will colors be manipulated? Saturation? From what I read and see in the figures, I would assume that the authors use rgb values and vary one of the three dimensions only? The following website gives an overview specifically related to PsychoPy and might be helpful in describing how stimuli are manipulated: <https://www.psychopy.org/general/colours.html> I was just thinking that it might be helpful to use DKL color space (monitor needs to be calibrated for this). Because in DKL space, color hues can be arranged in a color wheel and thus circularly. Just like the exemplars orientation. This would give the possibility to define color changes as everything within 0 deg or 90 deg along the color wheel. This would result in a nice consistency between the two feature dimensions.

- Other dimensions to choose stimuli: The authors would have to make sure that stimuli can only be chosen based on their orientation and color. From what it seems (Fig 2) participants

could also choose stimuli based on their location, e.g. some choice stimuli are closer to the initial exemplar position than others. One possible solution to this would be to organize objects in a circular arrangement around the exemplar location with equidistant locations and randomly rotate locations every trial minimize the potential implicit and explicit location strategies. Moreover, in Fig 2 it appears as if the different orientations are associated with a different aspect ratio of the rectangles. I assume that this is only due to the compressed figure?

--- CLARITY OF THE MANUSCRIPT ---

- Abstract: "we will manipulate memory retrieval by changes of the retention interval". It was not clear upon first reading whether the duration of the interval is changed or something else is happening in the interval.
- p.3: From the example at the beginning of the introduction it does not get clear why choice-consistency is considered a hallmark of rational value-based decision making as stated in the abstract and why it is worth investigating.
- P. 4, line 33-43 (Participants...research design): This part was unclear to me when reading it for the first time.
- P. 4, line 47: Please introduce multi-attribute visual choice shortly when the term is used for the first time.
- P. 5, line 37: it would be helpful if the authors could reiterate the two processes they are referring to.
- P. 6, line 39. What is the authors' framework? Fig. 1? Please specify
- P. 6, last paragraph. The hypothesis is clear, but I would recommend to nevertheless add one sentence like "That is, we expect a high choice consistency when... and a low choice consistency..." to further increase clarity
- P. 7., line 20: please briefly introduce the term out-of-sample data when using it for the first time.
- P. 8: I expected a paragraph about the setup and stimuli? Which monitor is used? How far away are subjects seated from the monitor? Given this distance, how big in degree visual angle do objects appear on the screen? Which characteristics do stimuli have? How are the stimuli controlled (PsychoPy)? And so on. I think adding this information would help a lot to understand the experiment.
- P. 9, lines 10-20: Related to the point above. It is hard to understand the stimuli and experimental procedure when it is only introduced as valued-based decision paradigm. It might have been clearer when stimuli and their characteristics were first introduced with the terms that would be normally used to describe a color-orientation memory experiment. I would recommend doing this first and then bridge the gap to how this can understood as a value-based decision paradigm.
- What is the task of participants? Please specify
- P. 10-11: Definitions 1-3: It would be helpful for the reader to give examples.
- Fig. 3: Having multiple lines would have helped me to better understand the experiment, the role of the budget and the resulting possible choice stimuli. Here, one choice stimulus is identical with regard to color. What about additional examples where it is identical with neither or with orientation? Or for different budgets? If this creates too much clutter, new panel(s) can be added. No changes are necessary if everything is sufficiently clear in the main text.

--- MINOR POINTS ---

- P. 4, l. 28: "older adults" is very unspecific. Please specify. E.g. (>60)
- P. 5, line 47-49: E Zarahn versus Eric Zarahn in in-text citation: I assume that the reference management system thinks these are two different authors and therefore highlights their first name?
- P. 8, line 28: Some guidelines including APA recommend to avoid one sentence paragraphs.

- P.9, line 5: Why does m need an index? When reading it first I understood it is always 100 and can thus be treated as a constant?
- Fig.1: Is there a reason for having two distinct choice nodes? Shouldn't this be the same behavior?
- Fig.2: Why does it need the square borders around the stimuli?
- Fig. 2: I would recommend to arrange these screenshots as a time line that states what is shown at which time during an individual trial.
- Fig.3: Please try to be more specific: What does relative mean?

Reviewer: 3

Comments to the Author(s)

The authors propose to use multi-attribute visual choice as a paradigm for assessing the influence of memory retrieval of goals on choice consistency. Specifically, they aim to experimentally test the influence of memory retrieval of exemplars on choice consistency in a visual choice paradigm.

As a disclaimer, I am not an expert on the content aspect of the paper and I have been invited to focus on the proposed Bayesian analyses which is the reason I will only comment on those.

Comments.

Overall, the proposed Bayesian analyses appear sound to me. However, I am not sure that the strategy of computing Bayes factors for only a training set and assessing out-of-sample predictive performance by performing a one-sided Bayesian t-test on the mean squared prediction error for the test set is ideal. I do not necessarily want to urge the authors to not conduct these analyses, however, I would like to ask them to consider the following additions:

1) I would like to see the Bayes factors that are computed based on the complete data set (i.e., test + training). These quantify the overall evidence in the data that will be collected and are of key interest.

2) An alternative approach to assessing out-of-sample predictive performance is to use the test set to update the priors for the parameters and then use these new priors (i.e., the posteriors based on the training set) to predict the test data using a Bayes factor approach. That is, the adequacy of these predictions can naturally be tested using Bayes factors (see Ly et al., 2019).

References.

Ly, A., Etz, A., Marsman, M., & Wagenmakers, E.-J. (2019). Replication Bayes factors from evidence updating. *Behavior research methods*, 51, 2498-2508.

Reviewer: 4

Comments to the Author(s)

See attached report.

Author's Response to Decision Letter for (RSOS-200308.R0)

See Appendix B.

RSOS-200308.R1 (Revision)

Review form: Reviewer 1

Do you have any ethical concerns with this paper?

No

Recommendation?

Accept with minor revision

Comments to the Author(s)

I only have one remaining comment related to my earlier comment on inclusion of a no-choice option. Not everyone will entirely buy your argument here so I suggest you mention something related to this, perhaps in the conclusion section, as a potential limitation and further area of research.

Review form: Reviewer 2

Do you have any ethical concerns with this paper?

No

Recommendation?

Accept with minor revision

Comments to the Author(s)

The authors have done a good job in addressing my previous concerns.

As a consequence, probably also due to my questions regarding color, the authors switched from using color to varying orientation in two dimensions. However, there is one potential new issue that arises from replacing the color manipulation with a second orientation manipulation.

As the authors state "Multi-attribute visual choice (MAVC) describes the comparative judgement of visual objects that are characterized by multiple attributes, e.g. orientation, color, shape."

In the present manuscript, strictly speaking, only one attribute is manipulated: orientation.

Physically the X and Y rotations might be independent dimension but perceptually they will most certainly not:

For example, here is a quote from the groundbreaking work on mental rotation by Shepard & Metzler (1971): "since they [participants] perceived the two-dimensional pictures as objects in three-dimensional space, they could imagine the rotation around whichever axis was required with equal ease"

As far as I understand the two dimensions should be (more or less) independent for a MAVC experiment. However, I am not completely sure whether this constitutes a potential problem for the present experiment or not. If it does, the authors might want to consider using a (more) independent dimension.

Shape and orientation might be problematic in combination (e.g. when one end of the shape dimension is a circle or sphere). For 2D objects, luminance (black to white) and orientation might go together. The same holds for luminance (or color) and shape. A nice shape manipulation

where shape is manipulated along a continuum from 0 to 1 can be found in Herwig & Schneider (2015); doi: 10.1111/nyas.12672.

Review form: Reviewer 4

Do you have any ethical concerns with this paper?

No

Recommendation?

Major revision

Comments to the Author(s)

I think the authors have done a great job in addressing almost all concerns of the reviewers.

The only point I think that it still has not been satisfactorily addressed, is my previous comment about the relation of the proposed task with the allocation task as used in economics. The CCEI measure that the authors will use to quantify violations of the GARP uses prices and quantities of goods (i.e. expenditures) to quantify how much each budget line has to be shifted in order two choices to satisfy the GARP. Quantities and prices are absent from the proposed task because subjects never trade off choices in terms of goods and prices and consequently their actions are not consequential in monetary terms. This isn't just about giving incentives to subjects but is more fundamental because utility functions are defined over goods, and prices along with income define the feasible choice set. I am unsure about what the utility function of a person trading off color and orientation would look like and how the subject would trade off these attributes since there is no explicit budget or preferences defined. The GARP provides a direct test of the utility model of preferences. What would a maximized utility function look like in the absence of goods and a budget? Is consistency with GARP a valid test in this case?

Decision letter (RSOS-200308.R1)

Dear Dr Nitsch,

On behalf of the Editors, I am pleased to inform you that your Manuscript RSOS-200308.R1 entitled "Influence of memory processes on choice consistency" has been accepted in principle for publication in Royal Society Open Science subject to minor revision in accordance with the referee and editor suggestions. Please find their comments at the end of this email.

The reviewers and handling editors have recommended publication, but also suggest some minor revisions to your manuscript. Therefore, I invite you to respond to the comments and revise your manuscript.

Please you submit the revised version of your manuscript within 7 days (i.e. by the 11-Aug-2020). If you do not think you will be able to meet this date please let me know immediately.

When submitting your revised manuscript, you will be able to respond to the comments made by the referees and you should upload a file "Response to Referees". You can use this to document any changes you make to the original manuscript. In order to expedite the processing of the revised manuscript, please be as specific as possible in your response to the referees.

Full author guidelines can be found here <https://royalsocietypublishing.org/rsos/registered-reports>.

on behalf of Professor Chris Chambers (Subject Editor, Royal Society Open Science)
openscience@royalsociety.org

Associate Editor Comments to Author (Professor Chris Chambers):

Associate Editor: 1

Comments to the Author:

Three of the reviewers have now assessed the revised manuscript, and the general consensus is that the manuscript is now close to being suitable for Stage 1 acceptance. As you will see, there remain some methodological issues to address (some prompted by new changes) either through further revision or clarification in the manuscript. Provided the authors are able to respond comprehensively to these points in a final revision, in-principle acceptance should be forthcoming without requiring further in-depth Stage 1 review.

Reviewer comments to Author:

Reviewer: 1

Comments to the Author(s)

I only have one remaining comment related to my earlier comment on inclusion of a no-choice option. Not everyone will entirely buy your argument here so I suggest you mention something related to this, perhaps in the conclusion section, as a potential limitation and further area of research.

Reviewer: 2

Comments to the Author(s)

The authors have done a good job in addressing my previous concerns.

As a consequence, probably also due to my questions regarding color, the authors switched from using color to varying orientation in two dimensions. However, there is one potential new issue that arises from replacing the color manipulation with a second orientation manipulation.

As the authors state “Multi-attribute visual choice (MAVC) describes the comparative judgement of visual objects that are characterized by multiple attributes, e.g. orientation, color, shape.”

In the present manuscript, strictly speaking, only one attribute is manipulated: orientation.

Physically the X and Y rotations might be independent dimension but perceptually they will most certainly not:

For example, here is a quote from the groundbreaking work on mental rotation by Shepard & Metzler (1971): “since they [participants] perceived the two-dimensional pictures as objects in three-dimensional space, they could imagine the rotation around whichever axis was required with equal ease”

As far as I understand the two dimensions should be (more or less) independent for a MAVC experiment. However, I am not completely sure whether this constitutes a potential problem for the present experiment or not. If it does, the authors might want to consider using a (more) independent dimension.

Shape and orientation might be problematic in combination (e.g. when one end of the shape dimension is a circle or sphere). For 2D objects, luminance (black to white) and orientation might go together. The same holds for luminance (or color) and shape. A nice shape manipulation where shape is manipulated along a continuum from 0 to 1 can be found in Herwig & Schneider (2015); doi: 10.1111/nyas.12672.

Reviewer: 4

Comments to the Author(s)

I think the authors have done a great job in addressing almost all concerns of the reviewers.

The only point I think that it still has not been satisfactorily addressed, is my previous comment about the relation of the proposed task with the allocation task as used in economics. The CCEI measure that the authors will use to quantify violations of the GARP uses prices and quantities of goods (i.e. expenditures) to quantify how much each budget line has to be shifted in order two choices to satisfy the GARP. Quantities and prices are absent from the proposed task because subjects never trade off choices in terms of goods and prices and consequently their actions are not consequential in monetary terms. This isn't just about giving incentives to subjects but is more fundamental because utility functions are defined over goods, and prices along with income define the feasible choice set. I am unsure about what the utility function of a person trading off color and orientation would look like and how the subject would trade off these attributes since there is no explicit budget or preferences defined. The GARP provides a direct test of the utility model of preferences. What would a maximized utility function look like in the absence of goods and a budget? Is consistency with GARP a valid test in this case?

Author's Response to Decision Letter for (RSOS-200308.R1)

See Appendix C.

Decision letter (RSOS-200308.R2)

Dear Dr Nitsch

On behalf of the Editor, I am pleased to inform you that your Stage 1 Registered Report RSOS-200308.R2 entitled "Influence of memory processes on choice consistency" has been accepted in principle for publication in Royal Society Open Science.

You may now progress to Stage 2 and complete the study as approved. Before commencing data collection we ask that you:

- 1) Update the journal office as to the anticipated completion date of your study.
- 2) Register your approved protocol on the Open Science Framework (<https://osf.io/>) or other recognised repository, either publicly or privately under embargo until submission of the Stage 2 manuscript. Please note that a time-stamped, independent registration of the protocol is mandatory under journal policy, and manuscripts that do not conform to this requirement cannot be considered at Stage 2. The protocol should be registered unchanged from its current approved state, with the time-stamp preceding implementation of the approved study design.

Following completion of your study, we invite you to resubmit your paper for peer review as a Stage 2 Registered Report. Please note that your manuscript can still be rejected for publication at Stage 2 if the Editors consider any of the following conditions to be met:

- The results were unable to test the authors' proposed hypotheses by failing to meet the approved outcome-neutral criteria.
- The authors altered the Introduction, rationale, or hypotheses, as approved in the Stage 1 submission.
- The authors failed to adhere closely to the registered experimental procedures. Please note that any deviations from the approved experimental procedures must be communicated to the editor immediately for approval, and prior to the completion of data collection. Failure to do so can result in revocation of in-principle acceptance and rejection at Stage 2 (see complete guidelines for further information).
- Any post-hoc (unregistered) analyses were either unjustified, insufficiently caveated, or overly dominant in shaping the authors' conclusions.
- The authors' conclusions were not justified given the data obtained.

We encourage you to read the complete guidelines for authors concerning Stage 2 submissions at <https://royalsocietypublishing.org/rsos/registered-reports#ReviewerGuideRegRep>. Please especially note the requirements for data sharing, reporting the URL of the independently registered protocol, and that withdrawing your manuscript will result in publication of a Withdrawn Registration.

Please note that Royal Society Open Science will introduce article processing charges for all new submissions received from 1 January 2018. Registered Reports submitted and accepted after this date will ONLY be subject to a charge if they subsequently progress to and are accepted as Stage 2 Registered Reports. If your manuscript is submitted and accepted for publication after 1 January 2018 (i.e. as a full Stage 2 Registered Report), you will be asked to pay the article processing charge, unless you request a waiver and this is approved by Royal Society Publishing.

You can find out more about the charges at <https://royalsocietypublishing.org/rsos/charges>. Should you have any queries, please contact openscience@royalsociety.org.

Once again, thank you for submitting your manuscript to Royal Society Open Science and we look forward to receiving your Stage 2 submission. If you have any questions at all, please do not hesitate to get in touch. We look forward to hearing from you shortly with the anticipated submission date for your stage two manuscript.

on behalf of Professor Chris Chambers (Registered Reports Editor, Royal Society Open Science)
openscience@royalsociety.org

Author's Response to Decision Letter for (RSOS-200308.R2)

See Appendix D.

RSOS-200308.R3 (Revision)

Review form: Reviewer 2

Do you have any ethical concerns with this paper?

Yes

Recommendation?

Accept with minor revision

Comments to the Author(s)

I could not see any deviations to the registered experimental procedure or analysis plan. The exploratory analyses are justified and appear sound. I just have a few minor comments that might help to improve the clarity of the manuscript.

Page 14, line 30: please introduce the abbreviation CCEI.

Page 14, line 36: It would have helped my understanding if the authors had briefly described how the ccei was computed.

Page 26, line 35: It might be worthwhile to briefly outline these concerns. For example, it was not clear to me whether these concerns also apply to the results reported in this paragraph and whether they can explain why other indices show a higher reliability.

Review form: Reviewer 4

Do you have any ethical concerns with this paper?

No

Recommendation?

Accept as is

Comments to the Author(s)

No additional comments.

Decision letter (RSOS-200308.R3)

Dear Dr Nitsch:

On behalf of the Editor, I am pleased to inform you that your Stage 2 Registered Report RSOS-200308.R3 entitled "Influence of memory processes on choice consistency" has been deemed suitable for publication in Royal Society Open Science subject to minor revision in accordance with the referee suggestions. Please find the referees' comments at the end of this email.

The reviewers and Subject Editor have recommended publication, but also suggest some minor revisions to your manuscript. Therefore, I invite you to respond to the comments and revise your manuscript.

Please also ensure that all the below editorial sections are included where appropriate -- if any section is not applicable to your manuscript, please can we ask you to nevertheless include the heading, but explicitly state that the heading is inapplicable. An example of these sections is attached with this email.

- **Ethics statement**

- **Data accessibility**

If you wish to submit your supporting data or code to Dryad (<http://datadryad.org/>), or modify your current submission to dryad, please use the following link:
[http://datadryad.org/submit?journalID=RSOS&manu=\(Document not available\)](http://datadryad.org/submit?journalID=RSOS&manu=(Document not available))

- **Competing interests**

- **Authors' contributions**

- **Acknowledgements**

- **Funding statement**

Because the schedule for publication is very tight, it is a condition of publication that you submit the revised version of your manuscript within 7 days (i.e. by the 24-Sep-2021). If you do not think you will be able to meet this date please let me know immediately.

1) A text file of the manuscript (tex, txt, rtf, docx or doc), references, tables (including captions) and figure captions. Do not upload a PDF as your "Main Document".

- 2) A separate electronic file of each figure (EPS or print-quality PDF preferred (either format should be produced directly from original creation package), or original software format)
- 3) Included a 100 word media summary of your paper when requested at submission. Please ensure you have entered correct contact details (email, institution and telephone) in your user account
- 4) Included the raw data to support the claims made in your paper. You can either include your data as electronic supplementary material or upload to a repository and include the relevant doi within your manuscript
- 5) All supplementary materials accompanying an accepted article will be treated as in their final form. Note that the Royal Society will neither edit nor typeset supplementary material and it will be hosted as provided. Please ensure that the supplementary material includes the paper details where possible (authors, article title, journal name).

on behalf of Professor Chris Chambers
(Registered Reports Editor, Royal Society Open Science)
openscience@royalsociety.org

Associate Editor Comments to Author (Professor Chris Chambers):

Associate Editor: 1

Comments to the Author:

Two of the original Stage 1 reviewers were available to evaluate the Stage 2 manuscript, and I extend my thanks to them for their prompt assessments during this difficult time. As you will see, both are positive about the submission with some useful suggestions for improving reproducibility and clarity of explanation. Provided you are able to respond thoroughly to these points in a revision, final acceptance should be forthcoming without requiring further in-depth review.

Comments to Author:

Reviewer: 2

Comments to the Author(s)

I could not see any deviations to the registered experimental procedure or analysis plan. The exploratory analyses are justified and appear sound. I just have a few minor comments that might help to improve the clarity of the manuscript.

Page 14, line 30: please introduce the abbreviation CCEI.

Page 14, line 36: It would have helped my understanding if the authors had briefly described how the ccei was computed.

Page 26, line 35: It might be worthwhile to briefly outline these concerns. For example, it was not clear to me whether these concerns also apply to the results reported in this paragraph and whether they can explain why other indices show a higher reliability.

Reviewer: 4

Comments to the Author(s)

I have no additional comments to make but I'd like to bring into the surface two points:

1) It is unclear which files should one run to replicate the analysis. I think the authors should include a readme file describing in detail the steps one should take to replicate everything and include more descriptive comments in their codes.

2) I hardly get Bayesian analysis so I think an expert reviewer should have a look.

The data surely allow the authors to test the proposed hypotheses, the authors follow the rationale as described in the Stage 1 submission and they adhered to the registered experimental procedures.

Author's Response to Decision Letter for (RSOS-200308.R3)

See Appendix E.

Decision letter (RSOS-200308.R4)

Dear Dr Nitsch:

It is a pleasure to accept your manuscript entitled "Influence of memory processes on choice consistency" in its current form for publication in Royal Society Open Science.

on behalf of Professor Chris Chambers (Subject Editor)
openscience@royalsociety.org

Appendix A

Review of ‘Influence of memory processes on choice consistency’ for Royal Society Open Science, RSOS-200308

The proposed Registered Report (RR) aims in examining the relationship of the strength of memory-based goal representations and choice consistency. The authors propose a novel methodology for directly measuring the process of how memory retrieval of goals and preferences affects choice consistency. They modify the budget allocation task of Choi et al. to a visual decision making task: subjects are shown an exemplar and will have to match the exemplar as close as possible by selecting through a set of choices that trade-off orientation and color. The slope of the budget implies different prices for the attributes ‘color similarity to exemplar’ and ‘orientation similarity to exemplar’. By varying prices (i.e. the slope of the budget line) and repeating the decision task, standard measures of choice consistency can be calculated. Moreover, the authors will manipulate memory representation strength of exemplars by changing the retention interval between exemplar presentation and choice. They expect an inverse relationship between retention and choice consistency.

Overall, I felt the research questions are valid and that the proposed hypotheses are plausible but I do have some further suggestions for improving the experimental design and describing things more clearly.

1. The standard budget allocation task of Choi et al. presents subjects with bundles of goods at given prices that are *consequential* to subjects. Subjects may end up with bundles of the goods that they have to pay for. Subjects’ implicit aim in this task is to maximize their utility. In the proposed RR, subjects will be instructed to trade-off orientation to color in order to match the exemplar as close as possible. Why would subjects follow this instruction? What is their motive to take this instruction at face value? What are the consequences for them if they don’t? Moreover, revealed preference theory is defined over quantities of goods and prices where the product of quantity and price is a cost for the person making the choice (as in Choi et al.). How does this relate to the modified decision task that you employ here? What is the cost of selecting one bundle over another? I think these issues need to be discussed in more depth in order to enhance the soundness of the proposed methodology.
2. The authors should consider including a manipulation check. Since the authors vary the retention interval between exemplar presentation and choice, it would further reinforce their research hypothesis if they could show that the retention interval has an effect on an independent task.
3. The authors should consider controlling for subjects’ memory ability by including an additional task that would measure ability.
4. The authors need a more robust justification of why they propose to do 40 trials of the decision task. The literature that applies the budget allocation task, selects the number of trials so that random choices cannot pass as consistent with revealed preferences theory (Bronar, 1987). One way to find out how many trials are optimal,

is to simulate random choices from a few thousands subjects and then calculate their CCEI. Random choices should be highly inconsistent with a higher number of trials, so that the less likely it is that random choice can pass as choice consistent.

5. Moreover, when designing the budget lines, the authors need to minimize the likelihood of cost efficient violations (see Murphy and Banerjee, 2015).
6. The CCEI measure that the authors plan to include is one of the measures that have been used to quantify choices consistent with GARP. There are a few other measures that they might want to consider like the Houtman-Maks index, the money-pump index, the Just-Noticeable difference and the minimum cost index (Dziewulski, 2019; Houtman and Maks, 1985; Echenique et al., 2011; Dean and Martin, 2016).
7. If I understand correctly each participant will be randomly assigned to a random retention interval from 0 to 30 and this interval will be kept constant throughout all trials. Please make this more clear in the description of the experiment.

References

- Bronars, Stephen G. 1987. "The Power of Nonparametric Tests of Preference Maximization." *Econometrica*, 55(3), 693-98.
- Dean, M. and D. Martin (2016). Measuring rationality with the minimum cost of revealed preference violations. *The Review of Economics and Statistics* 98 (3), 524-534.
- Dziewulski, P. (2019). Just-noticeable difference as a behavioural foundation of the critical cost-efficiency. Working Paper Series 0519, Department of Economics, University of Sussex Business School.
- Echenique, F., S. Lee, and M. Shum (2011). The money pump as a measure of revealed preference violations. *Journal of Political Economy* 119 (6), 1201-1223.
- Houtman, M. and J. Maks (1985). Determining all maximal data subsets consistent with revealed preference. *Kwantitatieve Methoden* 19, 89-104.
- Murphy, J.H., Banerjee, S. A caveat for the application of the critical cost efficiency index in induced budget experiments. *Exp Econ* 18, 356365 (2015).

Appendix B

Manuscript RSOS-200308

Response to Reviewers

Dear Professor Chambers,

thank you for giving us the opportunity to submit a revised draft of the manuscript “Influence of memory processes on choice consistency” as registered report S1 manuscript in *Royal Society Open Science*. We appreciate the time and effort that you and the reviewers dedicated to providing feedback on our manuscript and are grateful for the insightful comments on and valuable improvements to our paper. We have incorporated most of the suggestions made by the reviewers. Those changes are highlighted in yellow within the manuscript. Please see below, in blue, for a point-by-point response to the reviewers’ comments and concerns. All page numbers refer to the revised manuscript file with tracked changes.

Kind regards

Felix Jan Nitsch (on behalf of both authors)

Comments to Author:

Reviewer: 1

Comments to the Author(s)

I like the topic of the report and the experimental design is well-crafted. I have a few comments however that I would like the authors to consider before their data collection:

We thank you for your generally favourable assessment of our report and the constructive comments. Below we try to address all of your concerns.

1) Make the hypotheses more friendly to read and understand, and provide more discussion about the importance of each of the hypothesis and how results will contribute to the literature on choice consistency.

AUTHOR RESPONSE: We further clarified the hypotheses and added how evidence for or against each hypothesis will contribute to the literature.

“Based on theoretical considerations (Nosofsky, 1986, 2011; Wallenius et al., 2008) we propose that MAVC can serve as a model for value-based choice. In MAVC, we can experimentally manipulate memory representation strength of exemplars through changes of the retention interval between exemplar presentation and choice. This maps to a manipulation of the strength of memory-based goal representations in our framework (see figure 1). Revealed preference theory (Houthakker, 1950; Samuelson, 1938; Varian, 1982) can be used to analyze MAVC consistency without requiring assumptions about attribute weights or the parametric form of an integration function. Therefore, revealed preference theory can provide a general test of adherence to multi-attribute integration as formulated by the Generalized Context Model of categorization and multi-attribute utility theory. We propose the following hypothesis:

H1: As memory representations of exemplars are integral for MAVCs (Nosofsky, 2011), we expect choice consistency to decrease for longer retention intervals. That is, we expect an inverse relationship of retention interval between learning of the exemplar and choice, and choice consistency across multiple choices. Hence, we will provide experimental evidence on the role of memory representation strength of goals for choice consistency.

Previous research on the retention of information shows that forgetting curves are non-linear (Averell & Heathcote, 2011). As we expect choice consistency to be directly affected by the memory representation, we also expect the relationship of the retention interval and choice consistency to be non-linear.

H2: We expect choice consistency to decrease exponentially for longer retention intervals. That is, we expect an exponential model of the relationship of retention interval and choice consistency to be more strongly supported by the data than a null model (predicting a truncated normal distribution around the mean of the data). The evidence on H2 will help us to quantify the role of memory representation strength of goals for choice consistency beyond a directional prediction.

H3: In congruence with H2, we expect the exponential decrease of choice consistency for longer retention intervals to directly replicate in a new data set. This is important, as replicability is a minimal requirement on the meaningfulness of a psychological phenomenon.” (pages 7-8)

2) Page 8 – what does “participants must be healthy” really mean? Be more specific.

AUTHOR RESPONSE: We substituted “healthy” with a specific definition.

“Participants must be at least 17 years old, have normal or corrected vision, a good level of German, no neuropsychological or psychiatric diseases and give informed written consent.” (page 9)

3) My understanding is that the “trials” will consist of 5 rectangles and the subject will be asked to select one that looks most similar to the exemplar. My question is shouldn’t there be a “no-choice” option in each trial, especially since there can be trials where the choice set stimuli are quite different from the target stimuli? In many discrete choice experiments, researchers now tend to include a “no-choice or none of the above” option in each of the choice sets given to subjects.

AUTHOR RESPONSE: We appreciate the suggestion to include a no-choice option and agree that for some trials it will be difficult for the participants to make a similarity judgement. However, one core assumption of revealed preference theory is that there is a well-defined preference structure and this also holds for difficult decisions. Therefore, asking participants to make a choice for difficult decisions is part of a rigorous test of revealed preference choice consistency.

4) I wonder if results also be contingent upon the time that each exemplar is presented to subjects. In your present design, you plan to provide 5 seconds for this. Would results change if you gave them 10 seconds, for example?

AUTHOR RESPONSE: This is a very valid concern. Given the data we collected in our pilot experiment, we believe that a presentation time of 5 seconds is the best presentation time to test our retention interval hypotheses. However, we agree that it would be interesting to also consider the effect of the presentation time in future experiments.

“We found that precision increases with presentation time. In the context of our registered report, it is important that the memory representation strength freely varies among different retention intervals for a given presentation time. Descriptively, the variance of the absolute error in reconstruction is highest for a presentation time of 1 second. However, also the mean absolute error is highest for a presentation time of 1 second. A presentation time of 5 seconds represents a compromise with the second highest variance and second highest mean of the absolute error in reconstruction.” (page 24)

5) On top of page 11, it was mentioned that participants will be debriefed and reimbursed after they complete the second test block. This is vague and needs to be clarified more. For example, what would the debriefing entail? How are subjects being incentivized to put cognitive effort in their choices? Will one of the trials be randomly chosen as the binding trial?

AUTHOR RESPONSE: We further clarified the debriefing and reimbursement. Note, that in a perceptual choice paradigm entailing no correct choice it is difficult to incentivise task performance. Therefore, besides other reasons, we will include a control task with an objective performance measure which could be used to exclude disengaged participants. However, in our experience, serious disengagement has not been a problem with similar psychological experiments and both incentivized and non-incentivized tasks have borne similar results. Still, we are open to concrete suggestions on how to incentivize our task.

“After completion of the reconstruction task, participants will be debriefed about the goals of the study in written form and reimbursed via course credit.” (page 12)

Reviewer: 2

Comments to the Author(s)

Summary: Nitsch and Kalenschar aim to investigate the role of memory representations for choice consistency. As memory representations are typically unknown, they suggest that the underlying process can be investigated using a multi-attribute visual choice (MAVC) paradigm. In every trial of their version of the paradigm, participants will first see an exemplar stimulus. Stimuli will be bars differing in color and orientation. After a retention interval, participants will be confronted with five different bars of which participants have to select one. None of the stimuli will perfectly match with the exemplar but they will all constitute a trade-off between the two feature dimensions. The authors will manipulate the length of the retention interval and expect that this duration and choice consistency with regard to the two dimensions will be inversely related. That is, they expect that choice consistency will be high for short duration intervals and will decrease exponentially with increasing retention interval duration.

Evaluation: Overall, the manuscript investigates an interesting question and seems to be a promising and fruitful approach. However, I have a few recommendations regarding the clarity of the manuscript or questions that arise due to unclear or missing information.

Moreover, I have some concerns regarding the construction of stimuli and memory manipulation that I would recommend to address before collecting data. If the authors address these concerns, then the planned work will be scientifically strong and a meaningful contribution to the literature. I am looking forward to read the authors' reply and to see how the results turn out.

AUTHOR RESPONSE: We thank you for your in-depth review of our manuscript and appreciate the constructive recommendations which have led to a significant improvement of our registered report.

--- MEMORY INTERVAL & STIMULI ---

- **Efficiency of decay for a single item:** I doubt that information will necessarily decay in case of a single item with two informative features. According to the interpretative plan the absence of an inverse relationship between choice consistency and retention interval would “question the role of memory representation in [MAVC].” However, an alternative option is that this relationship exists but that the manipulation of retention interval was not successful. To overcome this, I would recommend to do a manipulation check. Here are some possible options that came to my mind:
- 1.) Introduce a manipulation check during the experiment.** For example, the authors might introduce a few memory trials at the end of every participant's session where they ask them to adjust the orientation and color of a previously seen exemplar. The time between showing the exemplar and the response screen should correspond to the two retention intervals tested. If the manipulation was successful, then precision should decay with the retention interval (i.e. larger orientation and hue errors). This procedure would come at a risk: If it turns out that the manipulation was not successful, then no conclusions could be drawn about the relationship between memory representation and MAVC.
 - 2.) Pilot the success of the memory manipulation before recording the actual experiment.** This pre-test could be done with a small additional sample that do a few trials each with e.g. 0, 15 and 30 s retention interval. When asking for color and orientation it would be important to vary the response order (due to response times adding to the retention interval) or to let participants judge/adjust both dimension simultaneously. Such a pilot experiment might turn out to be the safer option as it might prevent recording up to 500 participants with an unsuccessful manipulation. If the authors decide for a pilot experiment, and the manipulation turns out to be not successful, then the authors might want to bet their money on a different horse, e.g. ...
 - 3.) ...interference instead of decay:** Instead of varying the time between exemplar and choice, the authors might additionally (or instead) try to interfere with the memory representation, e.g. by showing a mask. The mask could consist of several colored lines in various orientations across the screen and the authors could vary the strength of the mask by varying its contrast. Other manipulations are of course possible. But this would only be necessary if the retention manipulation is not successful. Also in this case a manipulation check would be necessary.

AUTHOR RESPONSE: This is a valid concern which we addressing three-fold:

First, we have included a mask to the retention interval to increase possible effects on the exemplar memory representation. Note, that in response to a comment below (“- Color”),

we have exchanged the rectangle stimuli for cubes (for an explanation see our response to that comment).

“After presentation of the cube a mask of 10 similar cubes (with random X- and Y-orientations) will be presented to the participants for a short retention interval.” (page 9)

Second, we have conducted a pilot experiment to validate our retention interval in a similar control paradigm. Specifically, we asked participants to reconstruct the X- or Y-axis orientation of exemplar cubes. Similar to our planned study, we manipulated the retention interval (being either 1, 5, 10 or 30 seconds). We analysed the data using a Bayesian ANOVA-style model.

“We found that there was $BF_{10} = 1000 \pm 1.13\%$ times more evidence for the inclusion of the retention interval factor. We interpret this as definitive evidence. Inspection of the trend of means reveals that there is a positive relationship of retention interval and absolute error of reconstruction [...].” (page 23)

Third, we will also include this control paradigm in our main experiment as a manipulation check.

“After participants have completed the second test block, they will solve similar exemplar reconstruction task as a quality control and manipulation check. The first three screens of each trial will be equivalent to the procedure of the main task. Each trial will start with the presentation of a fixation cross. Participants will then be presented with an exemplar cube with a certain orientation along the X- and Y-axis for 5 seconds. Then, participants will be presented with a mask of 10 randomly oriented cubes (see figure 2) for a certain retention interval. Each participant is assigned a retention interval of either 1, 5, 10 or 30 seconds for the memory reconstruction task. After the retention interval, participants will again be presented with a single cube similar to the exemplar. The cube will randomly match the exemplar either regarding the X- or the Y-orientation, while the initial complementary orientation will be chosen uniform randomly. Participants then will have to turn the cube on the screen to match the exemplar regarding the complementary orientation using the arrow keys. Importantly, it will be unknown to the participants whether they will have to reconstruct the X- or Y-orientation both during the presentation time and the retention interval. Participants will solve 50 trials of the reconstruction task. Importantly, we will not use the results from the reconstruction task for our main analyses but as a manipulation check.” (pages 11-12)

- **Orientation: cardinal axes.** Related to my point above. People have a higher sensitivity to orientations along the cardinal axes compared to any other orientation. This would certainly also affect the memory representation. For example, a bar along the vertical could be remembered by remembering that it was vertical, whereas remembering a bar with an additional rotation of +10 deg would have to rely on some sort of visual memory. I would assume that the decay would only be successful (if at all) for orientations different than the cardinal axes. The authors might want to consider this when creating the exemplars and choice stimuli.

AUTHOR RESPONSE: We have excluded the cardinal axes from the set of possible orientations.

“The exemplar cube will have an orientation of 10, 75, 120, 185, 250 or 315 degrees on the X- and Y-Axis and an orientation of 0 degrees on the Z-Axis.” (page 9)

- **Color:** The manuscript is lacking some important information about how the color stimuli will be manipulated/created. Which color space do the authors use? Along which dimension will colors be manipulated? Saturation? From what I read and see in the figures, I would assume that the authors use rgb values and vary one of the three dimensions only? The following website gives an overview specifically related to PsychoPy and might be helpful in describing how stimuli are manipulated:

<https://www.psychopy.org/general/colours.html>

I was just thinking that it might be helpful to use DKL color space (monitor needs to be calibrated for this). Because in DKL space, color hues can be arranged in a color wheel and thus circularly. Just like the exemplars orientation. This would give the possibility to define color changes as everything within 0 deg or 90 deg along the color wheel. This would result in a nice consistency between the two feature dimensions.

AUTHOR RESPONSE: We have clarified the used color space in the manuscript. Further, we highly appreciate the suggestion to make both exemplar stimulus dimensions circular. However, our lab does not have the necessary expertise and technical devices in order to implement the DKL color space. We have, therefore, decided to take a different approach and overhaul our experimental stimuli. Instead of coloured 2d-rectangles we will now use 3d-cubes. The advantage 3d-objects is, that their orientation can be manipulated in up to 3 dimensions. For our experiment we have chosen to manipulate the orientation of 3d-cubes on the X- and Y-axes. Hence, both stimulus dimensions (X- and Y-orientation) are now entirely consistent.

- **Other dimensions to choose stimuli:** The authors would have to make sure that stimuli can only be chosen based on their orientation and color. From what it seems (Fig 2) participants could also choose stimuli based on their location, e.g. some choice stimuli are closer to the initial exemplar position than others. One possible solution to this would be to organize objects in a circular arrangement around the exemplar location with equidistant locations and randomly rotate locations every trial minimize the potential implicit and explicit location strategies.

Moreover, in Fig 2 it appears as if the different orientations are associated with a different aspect ratio of the rectangles. I assume that this is only due to the compressed figure?

AUTHOR RESPONSE: We appreciate the reviewer's suggestion. The choice stimuli now will be presented in 5 equidistant positions to the exemplar in random order. We overhauled figure 2 and fixed the aspect ratio problem (see response to the comment above).

--- CLARITY OF THE MANUSCRIPT ---

- **Abstract: "we will manipulate memory retrieval by changes of the retention interval". It was not clear upon first reading whether the duration of the interval is changed or something else is happening in the interval.**

AUTHOR RESPONSE: We clarified this in the manuscript.

"We will manipulate memory retrieval by varying the duration of the retention interval between exemplar presentation and choice." (page 2)

- **p.3: From the example at the beginning of the introduction it does not get clear why choice-consistency is considered a hallmark of rational value-based decision making as stated in the abstract and why it is worth investigating.**

AUTHOR RESPONSE: We further undermined the importance of choice consistency by outlining the implied money-pump effect and using a more implicative example.

"Imagine a stock trader who wants to trade stocks on two different days and plans to invest a starting capital of 600€. On the first day, shares of company A cost 200€ and shares of company B 150€. The stock trader buys 3 shares of company A and 0 shares of company B on the first day. On the second day, the share price of company A sinks to 150€ and the share price of company B rises to 200€. How should the stock trader respond to such a volatile stock market?

A naïve (and inconsistent) stock trader might be tempted to prematurely sell the shares of company A and instead invest into company B. However, this would incur sensitive losses to the trader (de facto 150€, a fourth of the starting capital). More importantly, continuously selling shares cheaper than buying them will inevitably lead to the loss of all capital and being driven out of the market (the so-called money pump phenomenon). Such investment behavior might, for example, arise from an inconsistent company value definition. In contrast, a consistent stock trader would base trading decisions on financial analysis, for example considering liquidity, book-to-market value, degree of state-ownership and past performance (Ng & Wu, 2006). This would result in a more robust value definition of company shares than the share price on a given day. Such a stock trader would, ideally, buy stocks at low prices and sell stocks for a profit, using the price volatility advantageously." (page 3)

- **P. 4, line 33-43 (Participants...research design): This part was unclear to me when reading it for the first time.**

AUTHOR RESPONSE: We tried to make this part clearer in the manuscript.

“Participants rated a catalog of food items on a Likert scale. Afterwards, they made repeated pair-wise choices between all possible pairs of food items from the catalog. Memory ability heterogeneity affected divergence of food ratings and actual choices. That is, participants with worse memory ability tended to more frequently choose items with a lower rating over items with a higher rating. However, unexpectedly, memory ability did not influence transitivity of choice itself. It is important to note, that the study by Levin et al. (2019) did not offer any direct measurements of choice-relevant memory retrieval and deploys a non-experimental research design.” (pages 4-5)

- **P. 4, line 47: Please introduce multi-attribute visual choice shortly when the term is used for the first time.**

AUTHOR RESPONSE: We now introduce multi-attribute visual choice shortly on first usage and additionally introduced the acronym suggested by the reviewer (MAVC).

“Multi-attribute visual choice (MAVC) describes the comparative judgement of visual objects that are characterized by multiple attributes, i.e. orientation, color, shape.” (page 5)

- **P. 5, line 37: it would be helpful if the authors could reiterate the two processes they are referring to.**

AUTHOR RESPONSE: Instead of referring to “two processes” we now reiterate both processes.

“We propose that the process of comparing choice options to performance goals in value-based choice and to exemplars in MAVC is, psychologically, sufficiently similar to use multi-attribute visual choice as a model for value-based choice.” (page 6)

- **P. 6, line 39. What is the authors’ framework? Fig. 1? Please specify**

AUTHOR RESPONSE: We included a reference to figure 1, which outlines the framework.

“This maps to a manipulation of the strength of memory-based goal representations in our framework (see figure 1).” (page 7)

- **P. 6, last paragraph. The hypothesis is clear, but I would recommend to nevertheless add one sentence like “That is, we expect a high choice consistency when... and a low choice consistency...” to further increase clarity**

AUTHOR RESPONSE: We further clarified the hypotheses and added how evidence for or against each hypothesis will contribute to the literature (see response to comment 1 of reviewer 1).

- **P. 7., line 20: please briefly introduce the term out-of-sample data when using it for the first time.**

AUTHOR RESPONSE: In line with our overhauled statistical analysis we now reframed H3 in terms of replication on a new dataset.

“H3: In congruence with H2, we expect that relative advantage in support by the data for the exponential model to replicate to a new data set. That is, we expect the exponential decrease of choice consistency for longer retention intervals to directly replicate in a new data set. This is important, as replicability is a minimal requirement on the meaningfulness of a psychological phenomenon.” (page 8)

- **P. 8: I expected a paragraph about the setup and stimuli? Which monitor is used? How far away are subjects seated from the monitor? Given this distance, how big in degree visual angle do objects appear on the screen? Which characteristics do stimuli have? How are the stimuli controlled (PsychoPy)? And so on. I think adding this information would help a lot to understand the experiment.**

AUTHOR RESPONSE: We added the requested information on monitor characteristics and distance to the monitor to the Procedure & Experimental Setup section.

“The experimental task will be presented with jsPsych (de Leeuw & Motz, 2016). All stimuli will be presented on a Lenovo ThinkPad T590 laptop. Subjects will be seated 30 cm away from the monitor in a dimly lighted room.” (page 12)

- **P. 9, lines 10-20: Related to the point above. It is hard to understand the stimuli and experimental procedure when it is only introduced as valued-based decision paradigm. It might have been clearer when stimuli and their characteristics were first introduced with the terms that would be normally used to describe a color-orientation memory experiment. I would recommend doing this first and then bridge the gap to how this can understood as a value-based decision paradigm.**

AUTHOR RESPONSE: See response to the comment below.

- **What is the task of participants? Please specify**

AUTHOR RESPONSE: We specified the task for the participant and strengthened the link to more common visual paradigms in the manuscript. Furthermore, we overhauled figure 2 as suggested above which should also contribute to a better understanding. Further we added an appendix with the instructions for the participants.

“The general notion of the task can be compared to that of delayed match-to-sample tasks (Habeck et al., 2004; Steffener et al., 2009, 2012; Zarahn, 2004; Zarahn et al., 2006, 2007) with the difference there is never a perfect match to the sample. Instead, the choice set stimuli represent a variable trade-off of orientation similarity to the exemplar regarding the X- and Y-axis. For example, a particular stimulus from the choice set might have a similar X-orientation but a different Y-orientation. Another stimulus might have a different X-orientation but a similar Y-orientation. Additionally, there can be trials where the choice set stimuli orientations resemble the exemplar orientation more closely and other trials where all choice set stimuli are quite differently oriented from the target stimuli. The task of the participants is, therefore, to mentally rotate each stimulus of the choice set until it matches the previously shown exemplar and evaluate which of the stimuli required the least mental rotation overall.” (page 10)

- **P. 10-11: Definitions 1-3: It would be helpful for the reader to give examples.**

AUTHOR RESPONSE: We further clarified the implications of the axiom and added a reference to figure 3.

“If the choice data pass Axiom 1, this means that choices are made as if integrated subjective similarity to the exemplar is a function of objective similarity along each attribute dimension (see figure 3). A simple example of such an integration function could be that subjective similarity is the weighed sum if the similarity along each attribute dimension. However, contrary to Nosofsky (1986), we do not need to make assumptions regarding the parametric form of such an integration function. Conversely, if the data do not pass Axiom 1 no GCM-style integration function of any monotonous concave specification can rationalize the data.” (pages 13-14)

- **Fig. 3: Having multiple lines would have helped me to better understand the experiment, the role of the budget and the resulting possible choice stimuli. Here, one choice stimulus is identical with regard to color. What about additional examples where it is identical with neither or with orientation? Or for different budgets? If this creates too much clutter, new panel(s) can be added. No changes are necessary if everything is sufficiently clear in the main text.**

AUTHOR RESPONSE: Figure 3 was generally overhauled to include multiple budget lines.

--- MINOR POINTS ---

- **P. 4, l. 28: “older adults” is very unspecific. Please specify. E.g. (>60)**

AUTHOR RESPONSE: We specified the sample characteristics of the study described.

“In a recent preregistered study, Levin et al. (2019) offered a trait heterogeneity based approach to the problem. The authors recruited people who were at least 65 years old to test for the effect of differences in memory abilities (measured by a cognitive assessment battery) on inconsistency in food choice” (page 4)

- **P. 5, line 47-49: E Zarahn versus Eric Zarahn in in-text citation: I assume that the reference management system thinks these are two different authors and therefore highlights their first name?**

AUTHOR RESPONSE: We fixed these issues.

- **P. 8, line 28: Some guidelines including APA recommend to avoid one sentence paragraphs.**

AUTHOR RESPONSE: We reformatted the document accordingly.

- **P.9, line 5: Why does m need an index? When reading it first I understood it is always 100 and can thus be treated as a constant?**

AUTHOR RESPONSE: We dropped the index for m whenever possible. Note, however, that revealed preference theory does not require m to be constant.

- **Fig.1: Is there a reason for having two distinct choice nodes? Shouldn't this be the same behavior?**

AUTHOR RESPONSE: We removed the duplicate choice node.

- **Fig.2: Why does it need the square borders around the stimuli?**

AUTHOR RESPONSE: We removed the square borders around the stimuli.

- **Fig. 2: I would recommend to arrange these screenshots as a time line that states what is shown at which time during an individual trial.**

AUTHOR RESPONSE: We overhauled figure 2 according to this suggestion (see response to comment above).

- **Fig.3: Please try to be more specific: What does relative mean?**

AUTHOR RESPONSE: Figure 3 was generally overhauled, including the figure description.

“Note. Choice sets for four different exemplars and sets of prices. Exemplars are shown for each example in the square in the upper right corner of each panel. From top left to bottom right: $p_1=(1,1)$, $p_2=(3,10)$, $p_3=(10,3)$, $p_4=(3,3)$. The size of the budget (set to $m=100$) relative to the prices determines how similar the choice set stimuli are oriented to the exemplar overall. Hence, the choice set stimuli in the top left panel are overall more similarly oriented to their respective exemplar than the choice set stimuli in the bottom right panel. The price ratio of the attributes determines the trade-off ratio of the X- and Y-orientation. Hence, the choice set stimuli in the top right panel are generally more similarly oriented to their respective exemplar along the Y-axis and less similarly oriented along the X-axis compared to the bottom left panel and vice versa. Axiomatic choice theory proposes that subjective similarity increases as a function of how far a choice object is located to the top right (indicated by the dashed line).” (page 32)

Reviewer: 3

Comments to the Author(s)

The authors propose to use multi-attribute visual choice as a paradigm for assessing the influence of memory retrieval of goals on choice consistency. Specifically, they aim to experimentally test the influence of memory retrieval of exemplars on choice consistency in a visual choice paradigm.

As a disclaimer, I am not an expert on the content aspect of the paper and I have been invited to focus on the proposed Bayesian analyses which is the reason I will only comment on those.

AUTHOR RESPONSE: We thank you for your expert assessment of, and very concrete suggestions to improve the statistical analysis of our registered report.

Comments.

Overall, the proposed Bayesian analyses appear sound to me. However, I am not sure that the strategy of computing Bayes factors for only a training set and assessing out-of-sample predictive performance by performing a one-sided Bayesian t-test on the mean squared prediction error for the test set is ideal. I do not necessarily want to urge the authors to not conduct these analyses, however, I would like to ask them to consider the following additions:

AUTHOR RESPONSE: Thank you for this generally positive evaluation of our statistical analysis. We appreciate your recommendations and try to incorporate them into our analysis plan.

1) I would like to see the Bayes factors that are computed based on the complete data set (i.e., test + training). These quantify the overall evidence in the data that will be collected and are of key interest.

AUTHOR RESPONSE: See response to the comment below.

2) An alternative approach to assessing out-of-sample predictive performance is to use the test set to update the priors for the parameters and then use these new priors (i.e., the posteriors based on the training set) to predict the test data using a Bayes factor approach. That is, the adequacy of these predictions can naturally be tested using Bayes factors (see Ly et al., 2019).

AUTHOR RESPONSE: We highly appreciate the input regarding the statistical analysis. We revised the hypotheses and analyses according to the replication Bayes factor method by Ly et al. (2019). In a nutshell, this approach reframes our train-test-set split as a study and its direct replication. We understand Mr. Gronaus interest in the Bayes factors for the overall dataset. We believe, however, that given the revised statistical analysis of our manuscript the overall Bayes Factors will mostly be of special interest in case of conflicting evidence for H2 and H3. Hence, we will consult the overall Bayes factors in case of conflicting evidence for H2 and H3.

“In order to test whether the relative advantage in support by the data for the exponential model in comparison to the null model replicates to a new data set, we will obtain the replication Bayes factor using the held out test set using the method described by Ly et al. (Ly et al., 2019). The replication Bayes factor is given by Bayes Factor for the coerced data set divided by the Bayes factor for the training set (obtained for H2).

$$BF_{10}(d_{test}|d_{train}) = \frac{BF_{10}(d_{test}, d_{train})}{BF_{10}(d_{train})}$$

This evidence updating method does not require approximations and is especially useful for complex models as in our application case.” (page 17)

“Should we find conflicting evidence for H2 and H3, we will use the Bayes factor for the complete dataset ($BF_{10}(d_{test}, d_{train})$) to guide our interpretation. The Bayesian model comparison using the complete dataset quantifies the evidence for or against each model in light of all data. We will use the same interpretation framework as before, which means that we consider a Bayes factor of $BF \geq 10$ as conclusive evidence.” (page 18)

References.

Ly, A., Etz, A., Marsman, M., & Wagenmakers, E.-J. (2019). Replication Bayes factors from evidence updating. *Behavior research methods*, 51, 2498-2508.

Reviewer: 4

Comments to the Author(s)

The proposed Registered Report (RR) aims in examining the relationship of the strength of memory-based goal representations and choice consistency. The authors propose a novel methodology for directly measuring the process of how memory retrieval of goals and preferences affects choice consistency. They modify the budget allocation task of Choi et al. to a visual decision making task: subjects are shown an exemplar and will have to match the exemplar as close as possible by selecting through a set of choices that trade-off orientation and color. The slope of the budget implies different prices for the attributes ‘color similarity to exemplar’ and ‘orientation similarity to exemplar’. By varying prices (i.e. the slope of the budget line) and repeating the decision task, standard measures of choice consistency can be calculated. Moreover, the authors will manipulate memory representation strength of exemplars by changing the retention interval between exemplar presentation and choice. They expect an inverse relationship between retention and choice consistency.

Overall, I felt the research questions are valid and that the proposed hypotheses are plausible but I do have some further suggestions for improving the experimental design and describing things more clearly.

AUTHOR RESPONSE: Thank you for generally affirming the validity of our research questions and plausibility of our hypotheses. Further, we thank you for your constructive suggestions to improve our design.

1. The standard budget allocation task of Choi et al. presents subjects with bundles of goods at given prices that are consequential to subjects. Subjects may end up with bundles of the goods that they have to pay for. Subjects’ implicit aim in this task is to maximize their utility. In the proposed RR, subjects will be instructed to trade-off orientation to color in order to match the exemplar as close as possible. Why would subjects follow this instruction? What is their motive to take this instruction at face value? What are the

consequences for them if they don't? Moreover, revealed preference theory is defined over quantities of goods and prices where the product of quantity and price is a cost for the person making the choice (as in Choi et al.). How does this relate to the modified decision task that you employ here? What is the cost of selecting one bundle over another? I think these issues need to be discussed in more depth in order to enhance the soundness of the proposed methodology.

AUTHOR RESPONSE: Note, that in a perceptual choice paradigm entailing no correct choice it is difficult to incentivise task performance. Therefore, besides other reasons, we will include a control task (see above) with an objective performance measure which could be used to exclude disengaged participants. However, in our experience, serious disengagement has not been a problem with similar psychological experiments and both incentivized and non-incentivized tasks have borne similar results. Thus, we believe that our participants are sufficiently motivated to perform the task, even without exogenous incentives. Still, we are open to concrete suggestions on how to incentivize our task.

Further, we clarified the concept of prices in our paradigm.

“Similarly, the prices and budgets in our multi-attribute choice task constrain the choice set of visual objects out of all possible visual objects characterized by specific attribute values (see figure 3). Given a fixed budget, the prices determine how much ‘similarity’ to the exemplar a participant can ‘purchase’ along a given orientation axis. The ‘cheaper’ a given dimension, the more similarity to the exemplar on that dimension a participant can afford.” (page 10)

2. The authors should consider including a manipulation check. Since the authors vary the retention interval between exemplar presentation and choice, it would further reinforce their research hypothesis if they could show that the retention interval has an effect on an independent task.

AUTHOR RESPONSE: This is a valid concern which we are addressing two-fold:

First, we have conducted a pilot experiment to validate our retention interval in a similar control paradigm. However instead of choice consistency we assessed exemplar reconstruction accuracy, which provides an objective measure of exemplar retention. Similar to our planned study, we manipulated the retention interval (being either 1, 5, 10 or 30 seconds). We analysed the data using a Bayesian ANOVA-style model.

“We found that there was $BF_{10} = 1000 \pm 1.13\%$ times more evidence for the inclusion of the retention interval factor. We interpret this as definitive evidence. Inspection of the trend of means reveals that there is a positive relationship of retention interval and absolute error of reconstruction [...].” (page 23)

Third, we will also include this control paradigm in our main experiment as a manipulation check. Note, that in response to another reviewer we have changed the stimulus dimensions from orientation and color to X- and Y-orientation in a 3d space (see response to reviewer 2 regarding “- color”).

“After participants have completed the second test block, they will solve a similar exemplar reconstruction task as a quality control and manipulation check. The first three screens of each trial will be equivalent to the procedure of the main task. Each trial will start with the presentation of a fixation cross. Next, participants will be presented with an exemplar cube with a certain orientation along the X- and Y-axis for 5 seconds. Then, participants will be presented with a mask of 10 randomly oriented cubes (see figure 2) for a certain retention interval. Each participant will be assigned a retention interval of either 1, 5, 10 or 30 seconds for the memory reconstruction task. After the retention interval, participants will again be presented with a single cube similar to the exemplar. The cube will randomly match the exemplar either regarding the X- or the Y-orientation, while the initial complementary orientation will be chosen uniform randomly. Participants then will have to turn the cube on the screen to match the exemplar regarding the complementary orientation using the arrow keys on the keyboard. Importantly, it will be unknown to the participants whether they will have to reconstruct the X- or Y-orientation both during the presentation time and the retention interval. Participants will solve 50 trials of the reconstruction task. Importantly, we will not use the results from the reconstruction task for our main analyses but as a manipulation check.” (pages 11-12)

3. The authors should consider controlling for subjects’ memory ability by including an additional task that would measure ability.

AUTHOR RESPONSE: While controlling for individual variety in memory ability is indeed interesting, we do not think that this is necessary given the homogeneity and size of our sample. However, the task which we will deploy as a manipulation check can serve as a task specific measurement of memory ability, which could be considered in secondary analyses.

4. The authors need a more robust justification of why they propose to do 40 trials of the decision task. The literature that applies the budget allocation task, selects the number of trials so that random choices cannot pass as consistent with revealed preferences theory (Bronar, 1987). One way to find out how many trials are optimal, is to simulate random choices from a few thousands subjects and then calculate their CCEI. Random choices should be highly inconsistent with a higher number of trials, so that the less likely it is that random choice can pass as choice consistent.

AUTHOR RESPONSE: We conducted a simulation study to generate a benchmark level of consistency, given uniform random choices, as suggested. We included this rationale as the justification of our design in the manuscript.

“In order to make meaningful statements about the influence of memory processes it is not only necessary to experimentally manipulate these memory processes with a sufficient effect size but also to measure choice consistency with sufficiently sensitive measure. The sensitivity of our behavioral task to detect violations of choice consistency can be approximated using a simulation study (Bronars, 1987). We simulated a dataset of 1.000 virtual participants that made uniform random choices from 20 choice sets constructed as specified for our experiment (see Procedure). Results showed that 99% of the virtual participants violated choice consistency at least once with a median CCEI of 0.389 (see figure 7).” (page 19)

5. Moreover, when designing the budget lines, the authors need to minimize the likelihood of cost efficient violations (see Murphy and Banerjee, 2015).

AUTHOR RESPONSE: We also addressed cost efficient violations in our simulation study.

“Note, that of all 1.000 virtual participants only a single one had a CCEI of 1. Importantly, this participant also did not violate the revealed preference axioms (0 inconsistent choices). We are, therefore, confident that our design also minimizes cost-efficient inconsistent choices which would undermine the sensitivity of the CCEI measure specifically (Murphy & Banerjee, 2015).” (page 36)

6. The CCEI measure that the authors plan to include is one of the measures that have been used to quantify choices consistent with GARP. There are a few other measures that they might want to consider like the Houtman-Maks index, the money-pump index, the Just-Noticeable difference and the minimum cost index (Dziewulski, 2019; Houtman and Maks, 1985; Echenique et al., 2011; Dean and Martin, 2016).

AUTHOR RESPONSE: While we agree that there are several other interesting indices of choice consistency, it is important to point out that the characteristics and concepts of these indices vary, sometimes substantially. As our computational model and analysis requires precise, quantitative predictions, with respect, we prefer, therefore, to restrain our preregistered analysis on the CCEI. However, we will attempt to conceptually replicate our findings using other indices as part of secondary analyses and will weigh the findings of these analyses into our final interpretation.

“Further, we plan to explore the robustness of our results using similar indices such as the money pump index (Echenique et al., 2011), the Houtman-Maks-Index (Heufer & Hjertstrand, 2015) and the minimum cost index (Dean & Martin, 2016). However, since all of these metrics measure slightly different constructs we will restrain our preregistered analysis to the critical cost efficiency index.” (page 14)

7. If I understand correctly each participant will be randomly assigned to a random retention interval from 0 to 30 and this interval will be kept constant throughout all trials. Please make this more clear in the description of the experiment.

AUTHOR RESPONSE: Each participant is assigned a random retention interval for each of the two task blocks. We clarified this in the manuscript.

“Then they solve two consecutive blocks of 20 trials each. For each test block, each participant is assigned a uniform-random retention interval between 0 and 30 seconds (please refer to paragraph “Floor and ceiling effects” below for discussion of the optimal interval length. In total, participants make 20 decisions each for two distinct retention intervals.” (page 11)

Appendix C

Manuscript RSOS-200308

Response to Reviewers

Dear Professor Chambers,

thank you for giving us another opportunity to submit an updated version of the manuscript “Influence of memory processes on choice consistency” with minor revisions as registered report S1 manuscript in *Royal Society Open Science*. We highly appreciate that three of four reviewers and yourself took the time and effort to evaluate the revision of our manuscript and provide another round of helpful and constructive criticism. We are confident to be able to address all concerns left in this second revision of our manuscript. In addition to the changes during the last round, we marked all new changes in green in the manuscript. Below you find our responses to each of the reviewers’ comments with the relevant parts of the manuscript that we changed in quotation.

Kind regards

Felix Jan Nitsch (on behalf of both authors)

Comments to Author:

Reviewer: 1

Comments to the Author(s)

I only have one remaining comment related to my earlier comment on inclusion of a no-choice option. Not everyone will entirely buy your argument here so I suggest you mention something related to this, perhaps in the conclusion section, as a potential limitation and further area of research.

We are glad that we were able to address all but one of the concerns in our last revision. We agree that our argument regarding the inclusion of a no-choice option might not convince all of our readers. As the reviewer suggested, we added this concern as a possible limitation and area of future research to our interpretation section.

“Further limitations

As one anonymous reviewer pointed out, our MAVC paradigm does not include a no-choice option. Intuitively, for some trials it will be difficult for participants to make a similarity judgement. However, we decided not to include a no-choice option in our paradigm as one core assumption of revealed preference theory is that there is a well-defined preference structure (Afriat, 1973) and this also holds for difficult decisions. Therefore, asking participants to make a choice for difficult decisions is part of a rigorous test of revealed preference choice consistency. Still, practically, this might introduce additional noise into the decision behavior of participants. While the current registered report cannot entirely address this aspect of insufficiently defined preferences, future research should provide both theoretical and empirical accounts on the role of non-decisions for choice consistency.” (page 21)

Reviewer: 2

Comments to the Author(s)

Summary: The authors have done a good job in addressing my previous concerns.

As a consequence, probably also due to my questions regarding color, the authors switched from using color to varying orientation in two dimensions. However, there is one potential new issue that arises from replacing the color manipulation with a second orientation manipulation.

As the authors state “Multi-attribute visual choice (MAVC) describes the comparative judgement of visual objects that are characterized by multiple attributes, e.g. orientation, color, shape.”

In the present manuscript, strictly speaking, only one attribute is manipulated: orientation. Physically the X and Y rotations might be independent dimension but perceptually they will most certainly not:

For example, here is a quote from the groundbreaking work on mental rotation by Shepard & Metzler (1971): “since they [participants] perceived the two-dimensional pictures as objects in three-dimensional space, they could imagine the rotation around whichever axis was required with equal ease”

As far as I understand the two dimensions should be (more or less) independent for a MAVC experiment. However, I am not completely sure whether this constitutes a potential problem for the present experiment or not. If it does, the authors might want to consider using a (more) independent dimension.

Shape and orientation might be problematic in combination (e.g. when one end of the shape dimension is a circle or sphere). For 2D objects, luminance (black to white) and orientation might go together. The same holds for luminance (or color) and shape. A nice shape manipulation where shape is manipulated along a continuum from 0 to 1 can be found in Herwig & Schneider (2015); doi: 10.1111/nyas.12672.

AUTHOR RESPONSE: We are happy that we were able to address the reviewer’s previous concerns and thank them again for an in-depth re-evaluation of our revised manuscript.

We agree that while X and Y rotations are physically independent, they might not be perceptually. Indeed, there is evidence that it depends on the participants’ strategy whether mental rotations are performed in a piece-wise or holistic fashion, which is subject to inter-individual variance (Heil & Jansen-Osmann, 2008). However, this poses no problem for our MAVC experiment as revealed preference choice consistency allows both independent (i.e. additive) and non-independent integration (any monotonous concave function of the attribute dimensions). This is non-trivial and we have clarified this in our manuscript.

“As one anonymous reviewer correctly pointed out, mental rotation may not necessarily be performed in an independent, piecewise fashion, but possibly also in a holistic mode (Shepard & Metzler, 1971), at least for some participants (Heil & Jansen-Osmann, 2008). We want to emphasize that any concave monotonous similarity function is consistent with revealed preference theory. Hence, an independent (i.e. additive) treatment of the two rotation axes is not required for our model.” (page 14)

Further, we appreciate the reviewer's additional suggestions for other visual attribute dimensions which can be utilized for MAVC experiments. We believe, that future research should perform similar experiments for various different attribute dimensions to further evaluate the domain generality of choice consistency.

Reviewer: 4

Comments to the Author(s)

I think the authors have done a great job in addressing almost all concerns of the reviewers.

The only point I think that it still has not been satisfactorily addressed, is my previous comment about the relation of the proposed task with the allocation task as used in economics. The CCEI measure that the authors will use to quantify violations of the GARP uses prices and quantities of goods (i.e. expenditures) to quantify how much each budget line has to be shifted in order two choices to satisfy the GARP. Quantities and prices are absent from the proposed task because subjects never trade off choices in terms of goods and prices and consequently their actions are not consequential in monetary terms. This isn't just about giving incentives to subjects but is more fundamental because utility functions are defined over goods, and prices along with income define the feasible choice set. I am unsure about what the utility function of a person trading off color and orientation would look like and how the subject would trade off these attributes since there is no explicit budget or preferences defined. The GARP provides a direct test of the utility model of preferences. What would a maximized utility function look like in the absence of goods and a budget? Is consistency with GARP a valid test in this case?

AUTHOR RESPONSE: We are glad that we were able to address almost all of the reviewer's concerns. With the further clarification of their previous comment, we understand that we have not sufficiently addressed the interpretability of GARP and, specifically, the CCEI in a perceptual task in our previous response letter.

To address this concern, we would like to offer a different interpretation of the CCEI which you have also referenced in your previous review. Dziwulski (2018) has shown that the CCEI can also be interpreted as "a measure of dissimilarity between alternatives that is sufficient for the agent to tell them apart". This notion of dissimilarity naturally extends to perceptual stimulus features. Preferences, in a utility model sense, hence can be translated to the integration mode of several perceptual features. Coincidentally, these integration modes are very similar to common economic utility functions down to their functional form (Nosofsky, 1986). And while the consequences in our experimental task (given that it is an abstracted laboratory paradigm) are non-obvious, there is a multitude of examples where the integration of perceptual features into compound similarity has strong consequences, e.g. eyewitness identification. Here, the maximized utility would coincide with maximized similarity in the identification of the most similar person in a line-up (assuming the perpetrator is present).

References

- Dziewulski, P. (2018). *Just-noticeable difference as a behavioural foundation of the critical cost-efficiency index*.
https://www.economics.ox.ac.uk/materials/working_papers/4603/848-dziewulski.pdf
- Heil, M., & Jansen-Osmann, P. (2008). Sex Differences in Mental Rotation with Polygons of Different Complexity: Do Men Utilize Holistic Processes whereas Women Prefer Piecemeal Ones? *Quarterly Journal of Experimental Psychology*, *61*(5), 683–689.
<https://doi.org/10.1080/17470210701822967>
- Nosofsky, R. M. (1986). Attention, Similarity, and the Identification-Categorization Relationship. *Journal of Experimental Psychology: General*, *115*(1), 39–57.

Appendix D

Heinrich Heine University Düsseldorf ☒ 40204 Düsseldorf
Prof. Tobias Kalenscher, Comparative Psychology

To the editors of Royal Society Open Science

Dear editors,

I am writing to submit the Stage 2 of the registered report "Influence of Memory Processes on Choice Consistency" by Tobias Kalenscher and myself.

The submitted document contains a results and discussion section. Table 1 (interpretation plan) from the main manuscript was included for convenience. The raw and processed data, the analysis code, and the approved stage 1 manuscript are available on the Open Science Framework (<https://osf.io/2vx36/>; link also on page 5 of the stage 2 document). No data for any pre-registered study other than pilot data included at Stage 1 was collected prior to the date of IPA.

We confirm that the completed experiment(s) have been executed and analysed in the manner originally approved with any unforeseen changes in those approved methods and analyses clearly noted. Specifically, neither the Introduction or Methods section were altered from the approved Stage 1 submission, and the stated hypotheses were not amended or appended.

The interpretation of the data strictly followed the preregistered analysis plan. Beyond the preregistered analyses we conducted two clearly marked and separately reported exploratory analyses, to further investigate the quality of our data (additional quality checks).

The authors report no competing interests.

We thank you for your time and look forward to your reply.
Sincerely,

Felix Jan Nitsch (on behalf of both authors)

Faculty of Mathematics
and Natural Sciences
Experimental Psychology

Comparative Psychology

Felix Jan Nitsch

PhD Student

Phone +49 211 81-12382

Felix.Nitsch@hhu.de

Düsseldorf, 18. Okt. 2021

Heinrich Heine University

Düsseldorf

Universitätsstraße 1

40225 Düsseldorf

Building 23.02

Level 00 Room 84

www.hhu.de

Formatted: Space Before: 0 pt, After: 12 pt, Line spacing: single, Tab stops: Not at 8.85 pi

Appendix E

Manuscript RSOS-200308

Response to Reviewers

Dear Professor Chambers,

we thank you for giving us the opportunity to submit an updated version of the manuscript “Influence of memory processes on choice consistency” with minor revisions as registered report S2 manuscript in *Royal Society Open Science*.

We were very pleased to hear that the reviewers confirmed that our work “adhered to the registered experimental procedures” and that our exploratory analyses “are justified and appear sound”.

Both reviewers provided constructive minor comments and insightful suggestions to strengthen the clarity and reproducibility of our manuscript. We thank them for their valuable feedback. We have given due consideration to the reviewers’ comments and have revised our manuscript in light of them. A point-by-point response is provided below. All changes to the previously submitted version of our manuscript are further highlighted in yellow in the attached revised manuscript. We believe that we have fully addressed all concerns raised by the reviewers and hope for a favourable decision. Thank you for your time, efforts, and consideration.

Kind regards

Felix Jan Nitsch (on behalf of both authors)

Comments to Author:

Reviewer: 2

Comments to the Author(s)

“I could not see any deviations to the registered experimental procedure or analysis plan. The exploratory analyses are justified and appear sound. I just have a few minor comments that might help to improve the clarity of the manuscript.”

AUTHOR RESPONSE: We thank the reviewer for this positive assessment of our stage 2 submission and for their helpful comments.

“Page 14, line 30: please introduce the abbreviation CCEI.”

AUTHOR RESPONSE: We thank the reviewer for spotting this negligence. We now introduce the abbreviation on first mention.

“For each test block and participant, we calculated the critical cost efficiency index (CCEI; Afriat, 1972, 1973; Varian, 1991).” (page 14)

“Page 14, line 36: It would have helped my understanding if the authors had briefly described how the ccei was computed.”

AUTHOR RESPONSE: We agree that a brief description of the computation of the CCEI will facilitate the clarity of the manuscript and have added it, therefore, at the suggested place.

“The CCEI denotes the “amount by which each budget constraint must be adjusted in order to remove all violations of GARP” (Choi et al., 2007, p. 1927). Computationally, the CCEI presents a relaxation of Axiom 1, so that only $x_i R x_j \rightarrow \neg(x_j p_j \times \text{CCEI} > x_i p_j) \forall i, j \in I$ must hold.” (page 14)

“Page 26, line 35: It might be worthwhile to briefly outline these concerns. For example, it was not clear to me whether these concerns also apply to the results reported in this paragraph and whether they can explain why other indices show a higher reliability.”

AUTHOR RESPONSE: We thank the reviewer for this insightful comment. We briefly outlined the abovementioned concerns and picked up the argument of the reviewer to revise the paragraph (changes highlighted in yellow).

“In an attempt to further understand the quality of our data beyond our preregistered quality controls, we conducted a descriptive test-retest reliability analysis of the CCEI for the training and test set. As we reported recently elsewhere (Nitsch et al., 2021), there are concerns regarding the measurement reliability of the CCEI, which is especially problematic for correlational designs such as the one of the current study (Hedge et al., 2018).

Specifically, tasks designed to show robust between-group effects and, thus, low between-subject variability in the outcome measure are at risk for showing low test-retest reliability. Another risk factor specific to the CCEI is that the measure is dependent only on the magnitude of the most severe violation (see Preprocessing) and, thus, vulnerable to outliers. Our results indicated essentially no reliability of the CCEI between training and test set in the current study ($r = -0.033$). This was not driven by the difference in retention intervals of both measurements, or by low between-subject variability of the measure (see figure 11, panel A and B). A similar result showed for the money pump index ($r = 0.023$). However, interestingly, the Houtman-Maks-Index and the Minimum-Cost-Index showed a much higher (albeit still poor) test-rest reliability (HMI: $r = 0.303$, MCI: $r = 0.353$; see figure 11, panels C-E), which might be attributed to less vulnerability to outliers.” (page 26)

Reviewer: 4

Comments to the Author(s)

“The data surely allow the authors to test the proposed hypotheses, the authors follow the rationale as described in the Stage 1 submission and they adhered to the registered experimental procedures.”

“I have no additional comments to make but I'd like to bring into the surface two points:”

AUTHOR RESPONSE: We appreciate this positive evaluation of our registered report and thank the reviewer for their helpful suggestions.

1) It is unclear which files should one run to replicate the analysis. I think the authors should include a readme file describing in detail the steps once should take to replicate everything and include more descriptive comments in their codes.

AUTHOR RESPONSE: We thank the reviewer for this great pointer. We have cleaned up the folder structure of the OSF repository and added a comprehensive README, detailing how to replicate all data generated figures and results in the manuscript: <https://osf.io/wd2mb/>.

2) I hardly get Bayesian analysis so I think an expert reviewer should have a look.

AUTHOR RESPONSE: We thank the reviewer for this honest assessment and have added a few more sentences of explanation to make our analysis more accessible to readers less familiar with Bayesian statistics.

“As outlined above, our statistical analyses use a Bayesian framework of inference, specifically the Bayes Factor approach to model comparison. Bayes factors express the *relative* degree of evidence for one model over another, that is the ratio of probabilities of observing the data under each model (Makowski et al., 2019).” (page 25)